# Development and external validation of prognostic models for COVID-19 to support risk stratification in secondary care

Nicola J Adderley ,[1] Thomas Taverner,[1] Malcolm James Price,[1,2] Christopher Sainsbury,[1,3] David Greenwood,[1] Joht Singh Chandan ,[1] Yemisi Takwoingi,[1,2] Rashan Haniffa,[4,5] Isaac Hosier,[1] Carly Welch,[6,7] Dhruv Parekh ,[6,7] Suzy Gallier,[7] Krishna Gokhale,[1] Alastair K Denniston,[6,8] Elizabeth Sapey ,[6,7] Krishnarajah Nirantharakumar[1,9]

For numbered affiliations see end of article.

**Correspondence to**
Dr Nicola J Adderley;
n.j.adderley@bham.ac.uk

## ABSTRACT

**Objectives** Existing UK prognostic models for patients admitted to the hospital with COVID-19 are limited by reliance on comorbidities, which are under-recorded in secondary care, and lack of imaging data among the candidate predictors. Our aims were to develop and externally validate novel prognostic models for adverse outcomes (death and intensive therapy unit (ITU) admission) in UK secondary care and externally validate the existing 4C score.

**Design** Candidate predictors included demographic variables, symptoms, physiological measures, imaging and laboratory tests. Final models used logistic regression with stepwise selection.

**Setting** Model development was performed in data from University Hospitals Birmingham (UHB). External validation was performed in the CovidCollab dataset.

**Participants** Patients with COVID-19 admitted to UHB January–August 2020 were included.

**Main outcome measures** Death and ITU admission within 28 days of admission.

**Results** 1040 patients with COVID-19 were included in the derivation cohort; 288 (28%) died and 183 (18%) were admitted to ITU within 28 days of admission. Area under the receiver operating characteristic curve (AUROC) for mortality was 0.791 (95% CI 0.761 to 0.822) in UHB and 0.767 (95% CI 0.754 to 0.780) in CovidCollab; AUROC for ITU admission was 0.906 (95% CI 0.883 to 0.929) in UHB and 0.811 (95% CI 0.795 to 0.828) in CovidCollab. Models showed good calibration. Addition of comorbidities to candidate predictors did not improve model performance. AUROC for the International Severe Acute Respiratory and Emerging Infection Consortium 4C score in the UHB dataset was 0.753 (95% CI 0.720 to 0.785).

**Conclusions** The novel prognostic models showed good discrimination and calibration in derivation and external validation datasets, and performed at least as well as the existing 4C score using only routinely collected patient information. The models can be integrated into electronic medical records systems to calculate each individual patient's probability of death or ITU admission at the time

## Strengths and limitations of this study

► The University Hospitals Birmingham (UHB) development dataset represents one of the largest and most ethnically diverse patient cohorts within the UK.

► As part of the UHB COVID-19 response, all admitted patients underwent a wide range of investigations to support international research efforts examining prognostic markers allowing assessment of a wide range of possible predictors (demographic variables, symptoms, physiological measures, imaging and laboratory test results) with low levels of missing data.

► A limitation of the study was that the overall sample size was relatively small compared with that of the International Severe Acute Respiratory and Emerging Infection Consortium study and was limited to one UK geographical location.

► In the external validation cohort, we were unable to examine all of the predictors included in the original full UHB model due to only a reduced set of candidate predictors being available in CovidCollab.

► It was not possible to carry out stratified analysis by ethnicity as the UHB dataset contained too few patients in many of the strata, and no ethnicity data were available in the CovidCollab dataset.

of hospital admission. Implementation of the models and clinical utility should be evaluated.

## BACKGROUND

The COVID-19 pandemic has placed exceptional strain on healthcare systems globally. Health systems, and especially critical care services, can be overwhelmed, given the number of patients and the duration and severity of their illness. A proportion of patients with COVID-19 can deteriorate rapidly. Clinicians need to differentiate between those with COVID-19 who are at

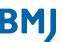

high risk of the most severe symptoms (requiring intensive care treatment/ventilation) or death, and those who can be considered at low risk and potentially managed in the community. Early identification of patients at highest risk of severe outcomes may provide opportunity to prioritise, intervene and improve outcomes.

Objective prognostic tools for patients with COVID-19, based on patients' initial characteristics, symptoms, biomarkers and imaging at the time of hospital admission, which can be used at or just after admission, and which can accurately discriminate between patients who will progress to more severe symptoms or death and those who will not, can be used by clinicians to triage and manage patients. This could potentially reduce time to appropriate interventions and improve patient outcomes.

A rapid systematic review has identified a number of prediction models developed for COVID-19, including prognostic models.[1] However, while these existing studies provided useful information on candidate predictors for further exploration, the review found substantial limitations: many models were developed exclusively in a Chinese population; many were at high risk of bias, particularly in terms of inclusion of non-representative control participants, inappropriate exclusion criteria and small sample sizes, leading to high risk of overfitting; and external validation was limited.[1] Other studies have evaluated existing early warning scores such as the National Early Warning Score, but with conflicting findings regarding their utility in predicting COVID-19 outcomes.[2 3]

More recent models have since been developed,[4 5] some of which overcome a number of these limitations, including the International Severe Acute Respiratory and Emerging Infection Consortium (ISARIC) model and corresponding (simplified) 4C score, which was developed in a UK secondary care population representing 260 hospitals in England, Scotland and Wales (the ISARIC dataset).[5] While the 4C score showed reasonable discrimination for mortality, there are some limitations, including a reliance on clinicians counting specific comorbidities, which may not be recorded at admission and which are known to be under-recorded in secondary care,[6] and an absence of imaging data among the candidate predictors.

### Aims and rationale

To date, there have been few prognostic models for patients admitted to the hospital with COVID-19 developed in a UK dataset. Furthermore, evaluation of the extent to which the inclusion of comorbidities, imaging and additional biomarkers improves model performance is required. It also remains to be determined whether updating the clinical parameters with evolving biomarkers improves prediction of the clinical course of patients as the disease evolves.

The overarching aim of this study was to develop prognostic models for patients admitted to the hospital with COVID-19 using routinely collected data at the point of admission, which can be used in a secondary care setting

to support clinical decision-making. Specific objectives were (1) to develop novel prognostic models for calculating predicted probability of adverse outcomes (death and intensive therapy unit (ITU) admission) at an individual patient level in a UK secondary care setting; (2) to externally validate these models in an international dataset (including data from UK hospitals); (3) to externally validate the existing UK ISARIC 4C score[5]; and (4) to compare performance of the newly developed models with the UK ISARIC 4C score. In addition, we developed daily models using time series data from the first 8 days from admission to explore changes in predictors over time.

## METHODS

### Data source

Data from University Hospitals Birmingham (UHB) NHS Foundation Trust were sourced via the PIONEER Health Data Research Hub for acute care and were used for model development and for external validation of the ISARIC 4C score. Data from patients with COVID-19 admitted to Queen Elizabeth Hospital, Birmingham (part of UHB), between 1 January 2020 and 16 August 2020 were included. Data included symptoms recorded at admission, comorbidities (from International Classification of Diseases, 10th revision (ICD-10) discharge codes), vital signs (eg, blood pressure and oxygen saturation), laboratory results (biochemistry, haematology, microbiology and pathology), imaging and outcomes (ITU admission and death).

External validation of the newly developed models was performed in the CovidCollab dataset. CovidCollab is an international project using routinely collected healthcare data to develop a better understanding of how best to treat and care for adults with COVID-19.[7 8] The dataset includes symptoms, comorbidities, vital signs, laboratory results, imaging findings and outcomes.

### Study population

Patients of all ages diagnosed with COVID-19 and hospitalised were included. Diagnosis was defined as a positive test result for SARS-CoV-2 from one or more reverse transcription PCR or transcription-mediated amplification tests. In the CovidCollab dataset, COVID-19 diagnosis was by either PCR or antibody test. Anonymised data for all patients with COVID-19 admitted to UHB during the study period were included. For CovidCollab, data collection was dependent on the specific processes within individual participating hospitals and the capacity of the data collector.[7]

### Study design

The study utilised retrospective cohort analyses; the index date (start of follow-up) was the hospital admission date. The study period was from 1 January 2020 to 12 September 2020 (the last admission date was 16 August to ensure a minimum of 28 days of follow-up).

## Outcomes

The primary outcome was death within 28 days of admission (in-hospital or post-discharge). The secondary outcome was ITU admission within 28 days of admission.

## Study follow-up

Participants were followed up from index (admission) date until the earliest of outcome date or study end (latest available data, 12 September 2020). Participants were censored 28 days after the index date. Participants admitted after 16 August 2020 (less than 28 days prior to the study end date) were excluded.

## Candidate predictor variables

Candidate predictors were selected a priori following a review of existing literature, discussion with clinical experts (specialists in acute care, critical care and geriatric medicine), and based on availability of variables routinely collected in secondary care/UHB. These included demographic variables, symptoms, comorbidities, physiological measures, imaging findings and laboratory test results. Comorbidities are not reliably and completely collected at admission, with the most complete hospital record of comorbidities usually being the discharge ICD-10 codes; therefore, the development and performance of models with and without comorbidity predictors were compared in order to explore the potential for developing models which would require no additional data collection (other than routinely collected data) at the point of admission.

## Model development

Models were trained using UHB data (patients admitted up to and including 16 August 2020). We used a multi-stage model building process that assessed the impact of a range of feature representation and modelling choices to select important candidate predictors. All analyses were performed in R.

Three sets of models were fitted which incorporated continuous variables in three different ways, to explore the impact of treating these variables as continuous or categorical, and also to explore the impact of different methods of handling missing data:

► As continuous numeric values, with missing values imputed ('continuous').
► As categorical values derived from the imputed continuous values ('categorical-imputed').
► In secondary analysis, as categorical values, using clinically meaningful categories and reference ranges, with missing indicators as a separate category ('categorical').

For the three ways of handling numerical features and missing variables mentioned previously, we fitted outcomes of death within 28 days and ITU admission (within 28 days) to candidate predictors using a range of models, which allowed both linear relationships and complex interactions between variables:

► Logistic regression with (1) all baseline parameters (demographic variables, symptoms, vital signs/ physiological measures and laboratory test results); (2) demographic variables only; and (3) all baseline parameters with the addition of recorded comorbidities (recorded up to the point of discharge).
► Logistic regression with stepwise Akaike information criterion (AIC) minimisation, both forward and backward.[9]
► Least absolute shrinkage and selection operator (LASSO, l1 penalised) logistic regression using all baseline parameters.
► Gradient boosted model (GBM) using all baseline parameters with default hyperparameter values of 150 trees, maximum interaction depth of 3, minimum of 10 observations in nodes and shrinkage of 0.1.[10]

Further information on handling of continuous variables is presented in online supplemental appendix 1.

For each of these four variable selection models, in order to reduce overfitting and selection bias, we internally validated using fivefold cross-validation (80/20 train/test split) to derive the candidate variable list. To avoid sensitivity to imputation, this cross-validation was repeated for each of the five multiple imputations.

Due to the relatively small number of outcome events (<300), we did not attempt to systematically look for interactions between multiple variables.

## Model performance

Model performance (discrimination) was assessed by calculating the area under the receiver operating characteristic curve (AUROC or C-statistic).[11] Calibration was assessed by plotting the observed probability of the outcome against predicted probability and by calculating the calibration slope and intercept. We also calculated sensitivity, specificity, positive predictive value (PPV) and negative predictive value (NPV) for the final models. For each feature set and each model, the final results for cross-validated (optimism-adjusted) AUROC and all other metrics (including calibration plots) were combined from all the multiple imputations of the dataset using Rubin's rules for the mean and CI (derived from the SD).[12]

## Missing data

Information on candidate predictors was collected at the point of admission; however, where information on physiological or laboratory measures was not available on the day of admission, measures recorded up to 72 hours after admission were used. Candidate predictors for which >40% of patients had missing data were excluded from the analysis. Further missing continuous variables (vital signs and laboratory tests) and symptoms were imputed using multiple imputation using chained equations (using the R 'mice' multiple imputation package). We performed five imputations and a maximum of 50 iterations.[13] Continuous variables were imputed with predictive mean matching, and categorical variables with logistic regression (logreg) or polytomous regression (polyreg). Input variables for the multiple imputation included all

available candidate predictor variables in the dataset; outcomes were not included in the imputation variables.

We also explored use of a missing category for missing test results. Absence of a record of a comorbidity was taken to indicate absence of the condition.

### External validation

To investigate the transferability of models, we performed external validation of logistic regression models derived from the UHB dataset in the CovidCollab dataset for predicting outcomes of 28-day mortality and ITU admission.

Not all candidate predictors were common to both datasets; therefore, new logistic regression models for death within 28 days and for ITU admission were refitted on the UHB data using only those variables also present in the CovidCollab data. We then performed an external validation of these UHB models in the CovidCollab dataset and ascertained the AUROC in both the UHB and Covid-Collab datasets. Based on model performance observed in the initial model derivation and in the interest of clinical utility, we used only categorical rather than continuous numerical variables, with imputed missing values (imputed prior to categorisation). To verify that predictors behaved similarly, we compared logistic coefficients from UHB to the same models fitted on the CovidCollab dataset. To account for sensitivity to missing values, we performed training and testing five times on fivefold multiple imputed datasets for both UHB and CovidCollab.

### External validation of ISARIC 4C score

A logistic regression using the 4C score was performed in the UHB dataset (following the same modelling methods used in the original ISARIC study). Model performance was assessed by calculating the AUROC and plotting calibration curves.

### Sensitivity analyses

Most patient records had some missing variables; we therefore performed a complete case analysis where we refitted the best forward stepwise selection model derived using the full set of UHB variables to complete case data, then data with ≤1, 2, 5 and 10 missing values, imputing missing values in the same way as previously mentioned, and examined AUROCs and logistic coefficients for stability.

In addition, we performed sensitivity analyses (1) within male and female strata by assessing performance (AUROC) of the final models in male and female patients separately; and (2) within age strata by assessing model performance in patients aged ≤60 and >60 years separately.

### Time series analysis

The UHB regression models used baseline measurement data collected on admission; where not available at admission, we accepted values up to 72 hours after admission. To investigate fine-grained temporal effects of data acquisition, we produced a series of separate logistic regression models using data collected at different time windows from

within 24 hours of admission up to within 7 days of admission, in 1-day increments, for the mortality outcome. Each dataset included only those patients eligible at the end of the window (not dead or discharged). This created eight different sets of predictors, including baseline variables of age, gender, symptoms and the time-sensitive variables of the latest physiological and laboratory measurements available.

For missing data, data were carried forward from the first observation (last observation carried forward (LOCF)) and fivefold multiple imputation was performed for missing data after LOCF was done, within each separate time-window dataset. Each model was trained and tested in fivefold cross validation, within each imputation, and AUROCs averaged using Rubin's rule. We compared the AUROCs for each of the eight models for predicting 28-day mortality from the time of admission and compared the logistic coefficients for the models. For additional insight into possible effects of changing measurements, we produced an additional logistic model for 28-day mortality to time-sensitive data collected within 4 days of admission, augmented with predictors indicating an increase or decrease in the category of each time-sensitive predictor relative to the reference category from 0 to 4 days, for example, whether temperature had crossed from below to above 37.8°C in that period.

### Patient and public involvement

We engaged with members of the PIONEER patient and public involvement group during development of the study protocol. We will further engage with this group, as well as other local and national patient and public involvement groups, in order to discuss dissemination of the findings and the best way to communicate these to patients and the public. We also consulted with several secondary care clinicians before and during the study to ensure that the tools developed meet the needs of clinicians. We have engaged with local NHS trusts to ensure that the algorithms developed are implemented/tested in a hospital setting.

## RESULTS

### Derivation cohort characteristics

A total of 1040 participants with COVID-19 admitted to UHB were included in the derivation cohort. A total of 288 (28%) died within 28 days of admission and 183 (18%) were admitted to ITU. Baseline characteristics are presented in table 1 (stratified by mortality outcome) and online supplemental table 1 (stratified by ITU admission). The mean (SD) age of participants was 68.2 (17.7) years; 57% (589) were male; and almost 90% had at least one comorbidity.

### Candidate predictors

After exclusion of seven candidate predictors with >40% missing data (D-dimer, ferritin, high-sensitivity troponin, fibrinogen, lactate dehydrogenase, vitamin D and

**Table 1** Baseline characteristics of participants admitted with COVID-19 in the derivation (UHB) and validation (CovidCollab) datasets

| | Development cohort (UHB) | | | External validation cohort (CovidCollab) | | |
|---|---|---|---|---|---|---|
| | Total | Alive | Died | Total | Alive | Died |
| **(A) Demographic characteristics, comorbidities and symptoms** | | | | | | |
| N | 1040 | 752 | 288 | 6099 | 4431 | 1668 |
| **Age category (years), n (%)** | | | | | | |
| <30 | 35 (3.4) | 35 (4.7) | 0 (0.0) | 125 (2.0) | 122 (2.8) | 3 (0.2) |
| 30–39 | 42 (4.0) | 39 (5.2) | 3 (1.0) | 270 (4.4) | 257 (5.8) | 13 (0.8) |
| 40–49 | 91 (8.8) | 87 (11.6) | 4 (1.4) | 497 (8.1) | 459 (10.4) | 38 (2.3) |
| 50–59 | 146 (14.0) | 123 (16.4) | 23 (8.0) | 793 (13.0) | 707 (16.0) | 86 (5.2) |
| 60–69 | 181 (17.4) | 143 (19.0) | 38 (13.2) | 944 (15.5) | 736 (16.6) | 208 (12.5) |
| 70–79 | 220 (21.2) | 147 (19.5) | 73 (25.3) | 1325 (21.7) | 891 (20.1) | 434 (26.0) |
| 80–89 | 214 (20.6) | 124 (16.5) | 90 (31.2) | 1571 (25.8) | 936 (21.1) | 635 (38.1) |
| ≥90 | 111 (10.7) | 54 (7.2) | 57 (19.8) | 574 (9.4) | 323 (7.3) | 251 (15.0) |
| Gender (male), n (%) | 589 (56.6) | 423 (56.2) | 166 (57.6) | 3361 (55.1) | 2342 (52.9) | 1019 (61.1) |
| **Ethnicity, n (%)** | | | | Not available | | |
| White | 590 (56.7) | 406 (54.0) | 184 (63.9) | | | |
| South Asian | 127 (12.2) | 95 (12.6) | 32 (11.1) | | | |
| Black | 68 (6.5) | 46 (6.1) | 22 (7.6) | | | |
| Other | 255 (24.5) | 205 (27.3) | 50 (17.4) | | | |
| **Comorbidities, n (%)** | | | | | | |
| Dementia | 373 (35.9) | 249 (33.1) | 124 (43.1) | 934 (15.3) | 539 (12.2) | 395 (23.7) |
| Cancer | 135 (13.0) | 90 (12.0) | 45 (15.6) | 649 (10.6) | 409 (9.2) | 240 (14.4) |
| Asthma | 165 (15.9) | 135 (18.0) | 30 (10.4) | 1579 (25.9)* | 1125 (25.4)* | 454 (27.2)* |
| Chronic obstructive pulmonary disease | 283 (27.2) | 211 (28.1) | 72 (25.0) | Not available | | |
| Sleep apnoea | 49 (4.7) | 36 (4.8) | 13 (4.5) | | | |
| Cardiovascular disease | 567 (54.5) | 366 (48.7) | 201 (69.8) | 3033 (49.7) | 1977 (44.6) | 1056 (63.3) |
| Hypertension | 661 (63.6) | 449 (59.7) | 212 (73.6) | Not available | | |
| Diabetes without complications | 258 (24.8) | 181 (24.1) | 77 (26.7) | 1794 (29.4) | 1229 (27.7) | 565 (33.9) |
| Diabetes with complications | 112 (10.8) | 76 (10.1) | 36 (12.5) | Not available | | |
| Peptic ulcer | 29 (2.8) | 25 (3.3) | 4 (1.4) | Not available | | |
| Liver disease | 71 (6.8) | 53 (7.0) | 18 (6.2) | Not available | | |
| Rheumatic/inflammatory disease | 51 (4.9) | 36 (4.8) | 15 (5.2) | Not available | | |
| Thyroid disorder | 107 (10.3) | 75 (10.0) | 32 (11.1) | Not available | | |

Continued

**Table 1** Continued

| | Development cohort (UHB) | | | External validation cohort (CovidCollab) | | |
|---|---|---|---|---|---|---|
| | Total | Alive | Died | Total | Alive | Died |
| **ISARIC comorbidity score** | | | | Not applicable | | |
| 0 | 111 (10.7) | 94 (12.5) | 17 (5.9) | | | |
| 1 | 234 (22.5) | 175 (23.3) | 59 (20.5) | | | |
| ≥2 | 695 (66.8) | 483 (64.2) | 212 (73.6) | | | |
| **Symptoms, n (%)** | | | | | | |
| Breathlessness | 559 (59.2) | 392 (56.6) | 167 (66.3) | Not available | | |
| Chest pain | 39 (4.1) | 33 (4.8) | 6 (2.4) | Not available | | |
| Cough | 538 (57.0) | 398 (57.5) | 140 (55.6) | 4259 (69.8) | 3110 (70.2) | 1149 (68.9) |
| Fever | 465 (49.3) | 339 (49.0) | 126 (50.0) | 3212 (52.7) | 2394 (54.0) | 818 (49.0) |
| Headache | 42 (4.4) | 37 (5.3) | 5 (2.0) | Not available | | |
| Malaise | 186 (19.7) | 147 (21.2) | 39 (15.5) | Not available | | |
| New-onset diarrhoea or vomiting | 56 (5.9) | 49 (7.1) | 7 (2.8) | Not available | | |
| Sputum | 84 (8.9) | 53 (7.7) | 31 (12.3) | Not available | | |
| Delirium | 80 (8.5) | 41 (5.9) | 39 (15.5) | 1160 (20.1) | 699 (16.7) | 461 (28.8) |
| **Outcomes** | | | | | | |
| Died within 28 days of admission | 288 (27.7) | 0 (0.0) | 288 (100.0) | 1668 (27.3) | 0 (0.0) | 1668 (100.0) |
| ITU admission within 28 days | 183 (17.6) | 132 (17.6) | 51 (17.7) | 722 (11.8) | 477 (10.8) | 245 (14.7) |
| **(B) Physiological measures and scores** | | | | | | |
| **BMI category, kg/m²** | | | | | | |
| Underweight (<18.5) | 27 (2.6) | 16 (2.1) | 11 (3.8) | 155 (2.5) | 112 (2.5) | 43 (2.6) |
| Normal weight (18.5–24.9) | 273 (26.2) | 179 (23.8) | 94 (32.6) | 1284 (21.1) | 940 (21.2) | 344 (20.6) |
| Overweight (25–29.9) | 341 (32.8) | 252 (33.5) | 89 (30.9) | 1247 (20.4) | 999 (22.5) | 248 (14.9) |
| Obese (≥30) | 366 (35.2) | 280 (37.2) | 86 (29.9) | 1180 (19.3) | 940 (21.2) | 240 (14.4) |
| Missing | 33 (3.2) | 25 (3.3) | 8 (2.8) | 2233 (36.6) | 1440 (32.5) | 793 (47.5) |
| **Systolic blood pressure (mm Hg), n (%)** | | | | | | |
| <140 | 688 (66.2) | 503 (66.9) | 185 (64.2) | 4051 (66.4) | 2936 (66.3) | 1115 (66.8) |
| ≥140 | 340 (32.7) | 249 (33.1) | 91 (31.6) | 1926 (31.6) | 1414 (31.9) | 512 (30.7) |
| Missing | 12 (1.2) | 0 (0.0) | 12 (4.2) | 122 (2.0) | 81 (1.8) | 41 (2.5) |
| **Diastolic blood pressure (mm Hg), n (%)** | | | | | | |
| <90 | 870 (83.7) | 636 (84.6) | 234 (81.2) | 5075 (83.2) | 3654 (82.5) | 1421 (85.2) |
| ≥90 | 158 (15.2) | 116 (15.4) | 42 (14.6) | 910 (14.9) | 701 (15.8) | 209 (12.5) |

Continued

**Table 1** Continued

| | Development cohort (UHB) | | | External validation cohort (CovidCollab) | | |
|---|---|---|---|---|---|---|
| | Total | Alive | Died | Total | Alive | Died |
| Missing | 12 (1.2) | 0 (0.0) | 12 (4.2) | 114 (1.9) | 76 (1.7) | 38 (2.3) |
| Temperature (degrees Celsius), n (%) | | | | | | |
| <37.8 | 851 (81.8) | 619 (82.3) | 232 (80.6) | 4164 (68.3) | 3025 (68.3) | 1139 (68.3) |
| ≥37.8 | 187 (18.0) | 133 (17.7) | 54 (18.8) | 1805 (29.6) | 1316 (29.7) | 489 (29.3) |
| Missing | 2 (0.2) | 0 (0.0) | 2 (0.7) | 130 (2.1) | 90 (2.0) | 40 (2.4) |
| Heart rate category (beats/min), n (%) | | | | | | |
| <80 | 288 (27.7) | 211 (28.1) | 77 (26.7) | 1654 (27.1) | 1190 (26.9) | 464 (27.8) |
| 80–99 | 441 (42.4) | 326 (43.4) | 115 (39.9) | 2400 (39.4) | 1794 (40.5) | 606 (36.3) |
| ≥100 | 309 (29.7) | 215 (28.6) | 94 (32.6) | 1935 (31.7) | 1370 (30.9) | 565 (33.9) |
| Missing | 2 (0.2) | 0 (0.0) | 2 (0.7) | 110 (1.8) | 77 (1.7) | 33 (2.0) |
| Respirations (breaths/min), n (%) | | | | | | |
| <20 | 450 (43.3) | 363 (48.3) | 87 (30.2) | 2249 (36.9) | 1769 (39.9) | 480 (28.8) |
| ≥20 | 573 (55.1) | 389 (51.7) | 184 (63.9) | 3659 (60.0) | 2522 (56.9) | 1137 (68.2) |
| Missing | 17 (1.6) | 0 (0.0) | 17 (5.9) | 191 (3.1) | 140 (3.2) | 51 (3.1) |
| Oxygen saturation (%), n (%) | | | | | | |
| <80 | 9 (0.9) | 4 (0.5) | 5 (1.7) | 158 (2.6) | 71 (1.6) | 87 (5.2) |
| 80–88 | 47 (4.5) | 17 (2.3) | 30 (10.4) | 443 (7.3) | 230 (5.2) | 213 (12.8) |
| 89–93 | 173 (16.6) | 110 (14.6) | 63 (21.9) | 1108 (18.2) | 775 (17.5) | 333 (20.0) |
| ≥94 | 809 (77.8) | 621 (82.6) | 188 (65.3) | 4281 (70.2) | 3284 (74.1) | 997 (59.8) |
| Missing | 2 (0.2) | 0 (0.0) | 2 (0.7) | 109 (1.8) | 71 (1.6) | 38 (2.3) |
| Partial pressure of $CO_2$ (kPa), n (%) | | | | | | |
| <4.67 | 184 (17.7) | 125 (16.6) | 59 (20.5) | 947 (15.5) | 627 (14.2) | 320 (19.2) |
| 4.67–6.3 | 380 (36.5) | 278 (37.0) | 102 (35.4) | 650 (10.7) | 467 (10.5) | 183 (11.0) |
| ≥6.4 | 176 (16.9) | 124 (16.5) | 52 (18.1) | 214 (3.5) | 128 (2.9) | 86 (5.2) |
| Missing | 300 (28.8) | 225 (29.9) | 75 (26.0) | 4288 (70.3) | 3209 (72.4) | 1079 (64.7) |
| Portable oxygen concentrator fraction of inspired oxygen (%), n (%) | | | | | | |
| ≤0.28 | 629 (60.5) | 458 (60.9) | 171 (59.4) | 3518 (57.7) | 2730 (61.6) | 788 (47.2) |
| 0.28–0.49 | 56 (5.4) | 35 (4.7) | 21 (7.3) | 1132 (18.6) | 794 (17.9) | 338 (20.3) |
| ≥0.5 | 80 (7.7) | 51 (6.8) | 29 (10.1) | 1003 (16.4) | 541 (12.2) | 462 (27.7) |
| Missing | 275 (26.4) | 208 (27.7) | 67 (23.3) | 446 (7.3) | 366 (8.3) | 80 (4.8) |
| Chest X-ray | | | | | | |

Continued

**Table 1** Continued

| | Development cohort (UHB) | | | External validation cohort (CovidCollab) | | |
|---|---|---|---|---|---|---|
| | **Total** | **Alive** | **Died** | **Total** | **Alive** | **Died** |
| Clear/unchanged | 210 (20.2) | 161 (21.4) | 49 (17.0) | 1604 (26.3) | 1225 (27.6) | 379 (22.7) |
| Local consolidation | 235 (22.6) | 169 (22.5) | 66 (22.9) | 3226 (52.9) | 2313 (52.2) | 913 (54.7) |
| Ground-glass opacity/bilateral infiltrates | 393 (37.8) | 273 (36.3) | 120 (41.7) | 637 (10.4) | 402 (9.1) | 235 (14.1) |
| Other/no firm diagnosis | 99 (9.5) | 65 (8.6) | 34 (11.8) | – | – | – |
| None performed/missing | 103 (9.9) | 84 (11.2) | 19 (6.6) | 632 (10.4) | 491 (11.1) | 141 (8.5) |
| Frailty score, n (%) | | | | | | |
| 1–3 | 376 (36.2) | 321 (42.7) | 55 (19.1) | 1451 (23.8) | 1326 (29.9) | 125 (7.5) |
| 4–6 | 277 (26.6) | 179 (23.8) | 98 (34.0) | 2079 (34.1) | 1539 (34.7) | 540 (32.4) |
| 7–9 | 119 (11.4) | 55 (7.3) | 64 (22.2) | 1911 (31.3) | 1146 (25.9) | 765 (45.9) |
| Missing | 268 (25.8) | 197 (26.2) | 71 (24.7) | 658 (10.8) | 420 (9.5) | 238 (14.3) |
| Glasgow Coma Scale score, n (%) | | | | | | |
| <15 | 274 (26.3) | 178 (23.7) | 96 (33.3) | 1314 (21.5) | 744 (16.8) | 570 (34.2) |
| 15 | 222 (21.3) | 186 (24.7) | 36 (12.5) | 4250 (69.7) | 3366 (76.0) | 884 (53.0) |
| Missing | 544 (52.3) | 388 (51.6) | 156 (54.2) | 535 (8.8) | 321 (7.2) | 214 (12.8) |
| (C) Laboratory test results | | | | | | |
| eGFR (mL/min), n (%) | | | | | | |
| <30 (stage 4 or above) | 200 (19.2) | 119 (15.8) | 81 (28.1) | 693 (11.4) | 364 (8.2) | 329 (19.7) |
| 30–59 (stage 3) | 216 (20.8) | 141 (18.8) | 75 (26.0) | 1362 (22.3) | 846 (19.1) | 516 (30.9) |
| 60–89 (stage 2) | 317 (30.5) | 246 (32.7) | 71 (24.7) | 1825 (29.9) | 1393 (31.4) | 432 (25.9) |
| >90 (normal or high) | 259 (24.9) | 215 (28.6) | 44 (15.3) | 1818 (29.8) | 1531 (34.6) | 287 (17.2) |
| Missing | 48 (4.6) | 31 (4.1) | 17 (5.9) | 401 (6.6) | 297 (6.7) | 104 (6.2) |
| pH, n (%) | | | | | | |
| <7.30 | 64 (6.2) | 39 (5.2) | 25 (8.7) | 416 (6.8) | 234 (5.3) | 182 (10.9) |
| 7.30–7.34 | 88 (8.5) | 64 (8.5) | 24 (8.3) | 530 (8.7) | 351 (7.9) | 179 (10.7) |
| 7.35–7.44 | 429 (41.2) | 313 (41.6) | 116 (40.3) | 2300 (37.7) | 1643 (37.1) | 657 (39.4) |
| ≥7.45 | 152 (14.6) | 107 (14.2) | 45 (15.6) | 960 (15.7) | 725 (16.4) | 235 (14.1) |
| Missing | 307 (29.5) | 229 (30.5) | 78 (27.1) | 1893 (31.0) | 1478 (33.4) | 415 (24.9) |
| Base excess (mmol/L), n (%) | | | | | | |
| <–2 | 202 (19.4) | 125 (16.6) | 77 (26.7) | 981 (16.1) | 560 (12.6) | 421 (25.2) |
| –2 to 2 | 349 (33.6) | 262 (34.8) | 87 (30.2) | 1996 (32.7) | 1491 (33.6) | 505 (30.3) |
| >2 | 182 (17.5) | 136 (18.1) | 46 (16.0) | 1077 (17.7) | 801 (18.1) | 276 (16.5) |

Continued

**Table 1** Continued

| | Development cohort (UHB) | | | External validation cohort (CovidCollab) | | |
|---|---|---|---|---|---|---|
| | Total | Alive | Died | Total | Alive | Died |
| Missing | 307 (29.5) | 229 (30.5) | 78 (27.1) | 2045 (33.5) | 1579 (35.6) | 466 (27.9) |
| Anion gap (mmol/L), n (%) | | | | Not available | | |
| 6–15 | 89 (8.6) | 62 (8.2) | 27 (9.4) | | | |
| ≥16 | 579 (55.7) | 409 (54.4) | 170 (59.0) | | | |
| Missing | 372 (35.8) | 281 (37.4) | 91 (31.6) | | | |
| White blood cell count (10$^9$/L), n (%) | | | | Not available | | |
| <3.9 | 79 (7.6) | 69 (9.2) | 10 (3.5) | | | |
| 3.9–10.8 | 696 (66.9) | 518 (68.9) | 178 (61.8) | | | |
| ≥10.9 | 215 (20.7) | 135 (18.0) | 80 (27.8) | | | |
| Missing | 50 (4.8) | 30 (4.0) | 20 (6.9) | | | |
| Platelets (10$^9$/L), n (%) | | | | Not available | | |
| <150 | 179 (17.2) | 121 (16.1) | 58 (20.1) | | | |
| 150–399 | 728 (70.0) | 538 (71.5) | 190 (66.0) | | | |
| ≥400 | 80 (7.7) | 62 (8.2) | 18 (6.2) | | | |
| Missing | 53 (5.1) | 31 (4.1) | 22 (7.6) | | | |
| Lymphocytes (10$^9$/L), n (%) | | | | | | |
| <1.5 | 801 (77.0) | 572 (76.1) | 229 (79.5) | 4684 (76.8) | 3349 (75.6) | 1335 (80.0) |
| ≥1.5 | 195 (18.8) | 154 (20.5) | 41 (14.2) | 1183 (19.4) | 929 (21.0) | 254 (15.2) |
| Missing | 44 (4.2) | 26 (3.5) | 18 (6.2) | 232 (3.8) | 153 (3.5) | 79 (4.7) |
| Neutrophil:lymphocyte ratio, n (%) | | | | | | |
| <2.21 | 94 (9.0) | 82 (10.9) | 12 (4.2) | 600 (9.8) | 509 (11.5) | 91 (5.5) |
| 2.21–4.82 | 282 (27.1) | 223 (29.7) | 59 (20.5) | 1635 (26.8) | 1341 (30.3) | 294 (17.6) |
| >4.82 | 620 (59.6) | 421 (56.0) | 199 (69.1) | 3387 (55.5) | 2265 (51.1) | 1122 (67.3) |
| Missing | 44 (4.2) | 26 (3.5) | 18 (6.2) | 477 (7.8) | 316 (7.1) | 161 (9.7) |
| Mean corpuscular volume (fL), n (%) | | | | Not available | | |
| <80 | 91 (8.8) | 69 (9.2) | 22 (7.6) | | | |
| 80–95 | 782 (75.2) | 578 (76.9) | 204 (70.8) | | | |
| ≥96 | 123 (11.8) | 79 (10.5) | 44 (15.3) | | | |
| Missing | 44 (4.2) | 26 (3.5) | 18 (6.2) | | | |
| Red cell distribution width (%), n (%) | | | | Not available | | |
| <11.5 | 13 (1.2) | 11 (1.5) | 2 (0.7) | | | |

Continued

**Table 1** Continued

| | Development cohort (UHB) | | | External validation cohort (CovidCollab) | | |
|---|---|---|---|---|---|---|
| | Total | Alive | Died | Total | Alive | Died |
| 11.5–15.4 | 742 (71.3) | 555 (73.8) | 187 (64.9) | | | |
| ≥15.5 | 240 (23.1) | 160 (21.3) | 80 (27.8) | | | |
| Missing | 45 (4.3) | 26 (3.5) | 19 (6.6) | | | |
| Monocytes ($10^9$/L), n (%) | | | | Not available | | |
| <0.2 | 71 (6.8) | 51 (6.8) | 20 (6.9) | | | |
| 0.2–0.8 | 731 (70.3) | 539 (71.7) | 192 (66.7) | | | |
| >0.8 | 194 (18.7) | 136 (18.1) | 58 (20.1) | | | |
| Missing | 44 (4.2) | 26 (3.5) | 18 (6.2) | | | |
| Eosinophils ($10^9$/L), n (%) | | | | Not available | | |
| ≤0.4 | 979 (94.1) | 710 (94.4) | 269 (93.4) | | | |
| >0.4 | 17 (1.6) | 16 (2.1) | 1 (0.3) | | | |
| Missing | 44 (4.2) | 26 (3.5) | 18 (6.2) | | | |
| Haemoglobin (g/L), n (%) | | | | | | |
| <115 | 310 (29.8) | 215 (28.6) | 95 (33.0) | 1454 (23.8) | 975 (22.0) | 479 (28.7) |
| 115–153 | 582 (56.0) | 434 (57.7) | 148 (51.4) | 3718 (61.0) | 2783 (62.8) | 935 (56.1) |
| ≥154 | 98 (9.4) | 73 (9.7) | 25 (8.7) | 557 (9.1) | 413 (9.3) | 144 (8.6) |
| Missing | 50 (4.8) | 30 (4.0) | 20 (6.9) | 370 (6.1) | 260 (5.9) | 110 (6.6) |
| Glucose (mmol/L), n (%) | | | | Not available | | |
| <7.8 | 585 (56.2) | 441 (58.6) | 144 (50.0) | | | |
| 7.8–8.4 | 76 (7.3) | 47 (6.2) | 29 (10.1) | | | |
| ≥8.5 | 295 (28.4) | 201 (26.7) | 94 (32.6) | | | |
| Missing | 84 (8.1) | 63 (8.4) | 21 (7.3) | | | |
| Bicarbonate (mmol/L), n (%) | | | | | | |
| <22 | 184 (17.7) | 118 (15.7) | 66 (22.9) | 996 (16.3) | 608 (13.7) | 388 (23.3) |
| 22–28 | 451 (43.4) | 339 (45.1) | 112 (38.9) | 2704 (44.3) | 1981 (44.7) | 723 (43.3) |
| ≥29 | 98 (9.4) | 66 (8.8) | 32 (11.1) | 437 (7.2) | 320 (7.2) | 117 (7.0) |
| Missing | 307 (29.5) | 229 (30.5) | 78 (27.1) | 1962 (32.2) | 1522 (34.3) | 440 (26.4) |
| C reactive protein (mg/L), n (%) | | | | | | |
| <10 | 84 (8.1) | 76 (10.1) | 8 (2.8) | 690 (11.3) | 616 (13.9) | 74 (4.4) |
| 10–99 | 406 (39.0) | 321 (42.7) | 85 (29.5) | 2735 (44.8) | 2101 (47.4) | 634 (38.0) |
| ≥100 | 483 (46.4) | 314 (41.8) | 169 (58.7) | 2213 (36.3) | 1381 (31.2) | 832 (49.9) |

Continued

**Table 1** Continued

| | Development cohort (UHB) | | | External validation cohort (CovidCollab) | | |
|---|---|---|---|---|---|---|
| | Total | Alive | Died | Total | Alive | Died |
| Missing | 67 (6.4) | 41 (5.5) | 26 (9.0) | 461 (7.6) | 333 (7.5) | 128 (7.7) |
| Albumin (g/L), n (%) | | | | Not available | | |
| <25 | 189 (18.2) | 123 (16.4) | 66 (22.9) | | | |
| 25–34 | 569 (54.7) | 408 (54.3) | 161 (55.9) | | | |
| ≥35 | 214 (20.6) | 176 (23.4) | 38 (13.2) | | | |
| Missing | 68 (6.5) | 45 (6.0) | 23 (8.0) | | | |
| Bilirubin (µmol/L), n (%) | | | | Not available | | |
| <21 | 822 (79.0) | 604 (80.3) | 218 (75.7) | | | |
| ≥21 | 151 (14.5) | 104 (13.8) | 47 (16.3) | | | |
| Missing | 67 (6.4) | 44 (5.9) | 23 (8.0) | | | |
| Alanine aminotransferase (U/L), n (%) | | | | | | |
| <55 | 837 (80.5) | 601 (79.9) | 236 (81.9) | 4126 (67.7) | 2986 (67.4) | 1140 (68.3) |
| ≥55 | 134 (12.9) | 106 (14.1) | 28 (9.7) | 777 (12.7) | 559 (12.6) | 218 (13.1) |
| Missing | 69 (6.6) | 45 (6.0) | 24 (8.3) | 1196 (19.6) | 886 (20.0) | 310 (18.6) |
| Alkaline phosphatase (U/L), n (%) | | | | Not available | | |
| <130 | 814 (78.3) | 605 (80.5) | 209 (72.6) | | | |
| ≥130 | 159 (15.3) | 103 (13.7) | 56 (19.4) | | | |
| Missing | 67 (6.4) | 44 (5.9) | 23 (8.0) | | | |
| Urea (mmol/L), n (%) | | | | | | |
| <7.8 | 567 (54.5) | 466 (62.0) | 101 (35.1) | 2585 (42.4) | 2122 (47.9) | 463 (27.8) |
| ≥7.8 | 429 (41.2) | 259 (34.4) | 170 (59.0) | 2409 (39.5) | 1399 (31.6) | 1010 (60.6) |
| Missing | 44 (4.2) | 27 (3.6) | 17 (5.9) | 1105 (18.1) | 910 (20.5) | 195 (11.7) |
| Potassium (mmol/L), n (%) | | | | Not available | | |
| 2.5–5.2 | 868 (83.5) | 644 (85.6) | 224 (77.8) | | | |
| ≥5.3 | 49 (4.7) | 29 (3.9) | 20 (6.9) | | | |
| Missing | 123 (11.8) | 79 (10.5) | 44 (15.3) | | | |
| Sodium (mmol/L), n (%) | | | | Not available | | |
| <133 | 146 (14.0) | 105 (14.0) | 41 (14.2) | | | |
| 133–144 | 746 (71.7) | 567 (75.4) | 179 (62.2) | | | |
| ≥145 | 104 (10.0) | 53 (7.0) | 51 (17.7) | | | |
| Missing | 44 (4.2) | 27 (3.6) | 17 (5.9) | | | |

Continued

**Table 1** Continued

| | Development cohort (UHB) | | | External validation cohort (CovidCollab) | | |
|---|---|---|---|---|---|---|
| | Total | Alive | Died | Total | Alive | Died |
| Corrected calcium (mmol/L), n (%) | | | | Not available | | |
| <2.2 | 142 (13.7) | 99 (13.2) | 43 (14.9) | | | |
| 2.2–2.5 | 767 (73.8) | 577 (76.7) | 190 (66.0) | | | |
| ≥2.6 | 38 (3.7) | 17 (2.3) | 21 (7.3) | | | |
| Missing | 93 (8.9) | 59 (7.8) | 34 (11.8) | | | |
| Lactate (U/L), n (%) | | | | | | |
| ≤2.2 | 490 (47.1) | 362 (48.1) | 128 (44.4) | 3091 (50.7) | 2327 (52.5) | 764 (45.8) |
| >2.2 | 192 (18.5) | 121 (16.1) | 71 (24.7) | 996 (16.3) | 558 (12.6) | 438 (26.3) |
| Missing | 358 (34.4) | 269 (35.8) | 89 (30.9) | 2012 (33.0) | 1546 (34.9) | 466 (27.9) |
| Haematocrit (L/L), n (%) | | | | Not available | | |
| <0.5 | 963 (92.6) | 707 (94.0) | 256 (88.9) | | | |
| ≥0.5 | 33 (3.2) | 19 (2.5) | 14 (4.9) | | | |
| Missing | 44 (4.2) | 26 (3.5) | 18 (6.2) | | | |

*In CovidCollab, asthma, chronic obstructive pulmonary disease and sleep apnoea were combined as 'respiratory diseases'. Data on the individual conditions were not available; therefore, the n (%) given is for all respiratory diseases.

BMI, body mass index; eGFR, estimated glomerular filtration rate; ISARIC, International Severe Acute Respiratory and Emerging Infection Consortium; ITU, intensive therapy unit; UHB, University Hospitals Birmingham.

haemoglobin A1c), 63 predictors were considered for inclusion in the models:

Demographic characteristics: age, gender and ethnicity.

Symptoms (binary, presence or absence of symptom at admission): breathlessness, chest pain, cough, fever, headache, malaise, new-onset diarrhoea or vomiting, sputum and delirium.

Physiological measures and vital signs: body mass index (BMI, kg/m$^2$), systolic blood pressure (mm Hg), diastolic blood pressure (mm Hg), temperature (degrees Celsius), heart rate (beats/min), respiratory rate (breaths/min), oxygen saturation (%), partial pressure of $CO_2$ (kPa) and portable oxygen concentrator fraction of inspired oxygen ($FiO_2$, %).

Imaging: chest X-ray finding (categorised as clear/unchanged, local consolidation, ground-glass opacity/bilateral infiltrates, other/no firm diagnosis, none performed/missing).

Scores: frailty score (Rockwood Clinical Frailty Scale)[14]; Glasgow Coma Scale score[15 16];

laboratory test results: estimated glomerular filtration rate (eGFR, ml/min), pH (%), base excess (mmol/L), anion gap (mmol/L), white blood cell (WBC) count ($10^9$/L), platelets ($10^9$/L), lymphocytes ($10^9$/L), neutrophil:lymphocyte ratio, mean corpuscular volume (fL), red cell distribution width (%), monocytes ($10^9$/L), eosinophils ($10^9$/L), haemoglobin (g/L), glucose (mmol/L), bicarbonate (mmol/L), C reactive protein (mg/L), albumin (g/L), bilirubin (μmol/L), alanine aminotransferase (U/l), alkaline phosphatase (U/l), urea (mmol/l), potassium (mmol/l), sodium (mmol/l), corrected calcium (mmol/l), lactate (U/l) and haematocrit (l/l);

Comorbidities (binary, presence or absence of record in discharge ICD-10 codes): dementia, cancer, asthma, chronic obstructive pulmonary disease, sleep apnoea, cardiovascular disease, hypertension, diabetes without complications, diabetes with complications, peptic ulcer, liver disease, rheumatic/inflammatory disease, thyroid disorder.

## Mortality outcome (28 days): UHB model and predictive performance

Area under the ROC curve values for each of the logistic, LASSO and GBM models, treating continuous variables in one of three ways (as continuous variables with imputed missing values; as clinically meaningful categorical variables with imputed missing values; and as categorical variables with missing categories), are presented in online supplemental table 2.

The final model selected was a logistic regression using stepwise selection of variables with categorisation of continuous variables (with imputed missing values). The final 18 categorical predictors included in the model were: age, breathlessness, sputum, systolic blood pressure, temperature, respiratory rate, oxygen saturation, $FiO_2$, alkaline phosphatase, C-reactive protein, corrected calcium, eosinophils, glucose, pH, urea, WBC count, platelets and frailty score.

AUROC for the UHB cross-validated model was 0.779 (95 % CI 0.744 to 0.813) (table 2). At a 20% predicted probability of mortality, sensitivity was 83% (95% CI 81% to 85%); specificity was 58% (95% CI 55% to 61%); positive predictive value was 43% (95% CI 41% to 46%); and negative predictive was 90% (95% CI 88% to 91%)

**Table 2** AUROCs, calibration slopes and calibration intercepts for models developed in UHB data (full (UHB) and reduced (UHB-R) datasets) and externally validated in CovidCollab data, and for external validation of the ISARIC 4C score

| Dataset | Outcome | AUROC (95% CI) | Calibration (95% CI) | |
|---|---|---|---|---|
| | | | Slope | Intercept |
| Model development* | | | | |
| UHB | Mortality | 0.779 (0.744 to 0.813) | 0.79 (0.64 to 0.94) | 0.06 (−0.02 to 0.14) |
| UHB | ITU admission | 0.893 (0.864 to 0.922) | 0.91 (0.80 to 1.01) | 0.01 (−0.05 to 0.07) |
| UHB-R† | Mortality | 0.791 (0.761 to 0.822) | 0.89 (0.81 to 0.97) | 0.03 (−0.01 to 0.07) |
| UHB-R† | ITU admission | 0.906 (0.883 to 0.929) | 0.94 (0.84 to 1.04) | 0.00 (−0.05 to 0.06) |
| External validation of new model* | | | | |
| CovidCollab | Mortality | 0.767 (0.754 to 0.780) | 0.85 (0.75 to 0.94) | 0.04 (−0.01 to 0.09) |
| CovidCollab | ITU admission | 0.811 (0.795 to 0.828) | 0.95 (0.82 to 1.08) | 0.00 (−0.07 to 0.07) |
| External validation of ISARIC 4C score in UHB data | | | | |
| UHB—4C score | Mortality | 0.753 (0.720 to 0.785) | 0.99 (0.85 to 1.12) | 0.00 (−0.06 to 0.06) |

*Models derived using logistic regression with stepwise selection of candidate predictors and categorisation of continuous variables into clinically meaningful categories (after imputing missing data).
†Not all variables included in the full UHB model were available in the CovidCollab dataset. Therefore, revised (reduced) models were developed in UHB data using a subset of the candidate predictors common to both the UHB and CovidCollab datasets (UHB-R), and these were then externally validated in the CovidCollab dataset.
AUROC, area under the receiver operating characteristic curve; ISARIC, International Severe Acute Respiratory and Emerging Infection Consortium; UHB, University Hospitals Birmingham; UHB-R, University Hospitals Birmingham reduced model.

**Table 3A** Sensitivity, specificity, PPV and NPV for mortality at 28 days after admission (University Hospitals Birmingham derivation dataset)

| Predicted probability (%) | TP | FP | TN | FN | Sensitivity (95% CI) | Specificity (95% CI) | PPV (95% CI) | NPV (95% CI) |
|---|---|---|---|---|---|---|---|---|
| 10 | 270 | 477 | 275 | 18 | 0.938 (0.924 to 0.953) | 0.365 (0.346 to 0.385) | 0.362 (0.352 to 0.371) | 0.939 (0.924 to 0.954) |
| 20 | 238 | 315 | 437 | 50 | 0.828 (0.806 to 0.850) | 0.581 (0.547 to 0.614) | 0.431 (0.405 to 0.456) | 0.898 (0.881 to 0.915) |
| 30 | 194 | 210 | 542 | 94 | 0.672 (0.646 to 0.699) | 0.721 (0.696 to 0.746) | 0.480 (0.458 to 0.502) | 0.852 (0.841 to 0.862) |
| 40 | 156 | 130 | 622 | 132 | 0.540 (0.515 to 0.565) | 0.827 (0.813 to 0.842) | 0.545 (0.523 to 0.567) | 0.825 (0.817 to 0.832) |
| 50 | 116 | 74 | 678 | 172 | 0.404 (0.366 to 0.442) | 0.902 (0.888 to 0.916) | 0.613 (0.578 to 0.648) | 0.798 (0.788 to 0.808) |
| 60 | 82 | 40 | 712 | 206 | 0.284 (0.250 to 0.318) | 0.947 (0.940 to 0.954) | 0.673 (0.641 to 0.704) | 0.775 (0.768 to 0.783) |
| 70 | 50 | 21 | 731 | 238 | 0.174 (0.147 to 0.200) | 0.972 (0.966 to 0.978) | 0.706 (0.671 to 0.741) | 0.754 (0.749 to 0.760) |
| 80 | 24 | 11 | 741 | 264 | 0.083 (0.073 to 0.094) | 0.986 (0.979 to 0.992) | 0.691 (0.576 to 0.807) | 0.737 (0.734 to 0.740) |
| 90 | 8 | 2 | 750 | 280 | 0.026 (0.015 to 0.038) | 0.997 (0.996 to 0.998) | 0.772 (0.663 to 0.881) | 0.728 (0.726 to 0.730) |

FN, false negative; FP, false positive; NPV, negative predictive value; PPV, positive predictive value; TN, true negative; TP, true positive.

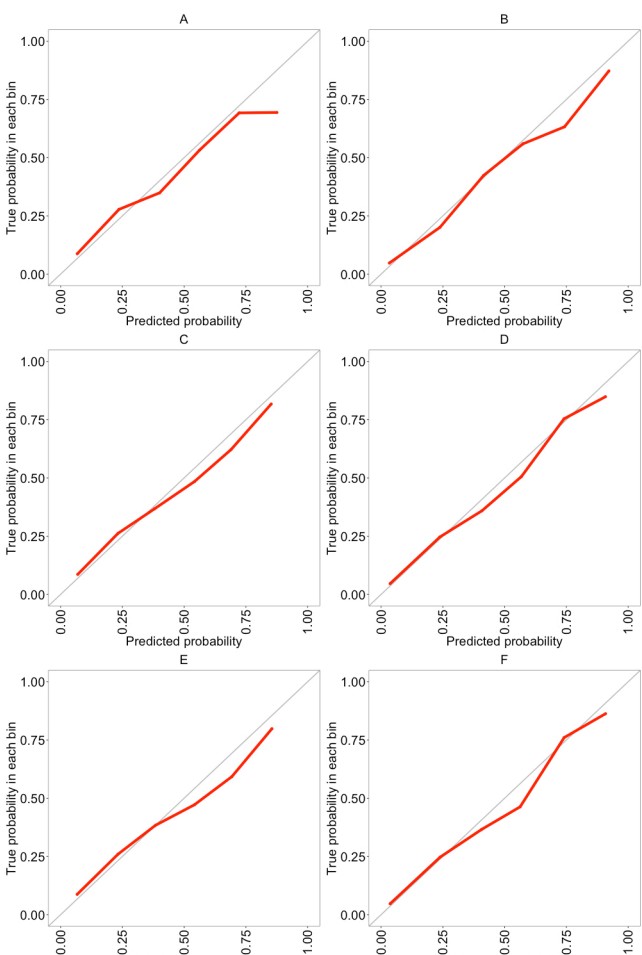

**Figure 1** Calibration plots (observed probability (y-axis) against predicted probability (x-axis)): (A) UHB derivation dataset for mortality outcome, (B) UHB derivation dataset for ITU admission outcome, (C) UHB-R derivation/train dataset reduced model for mortality outcome, (D) UHB-R derivation/train dataset reduced model for ITU admission outcome, (E) CovidCollab external validation dataset reduced model for mortality outcome, (F) CovidCollab external validation dataset reduced model for ITU admission outcome. ITU, intensive therapy unit; UHB, University Hospitals Birmingham; UHB-R, University Hospitals Birmingham reduced.

(table 3A). Calibration was very good at low to medium predicted probabilities but was poorer at very high predicted probabilities; a calibration plot is shown in figure 1A; the calibration slope was 0.79 (95% CI 0.64 to 0.94) (table 2). Model coefficients (and model equation) are presented in online supplemental table 3.

Addition of comorbidities to the candidate predictors included in the model did not improve performance of the model (online supplemental table 2). Since comorbidities are known to be under-reported during acute presentations,[6] and they offered no improvement on model performance, models without comorbidities were preferred.

### ITU admission: UHB model and predictive performance

Area under the ROC curve values for each of the models performed are presented in online supplemental table 4.

The final model selected was a logistic regression using stepwise selection of variables with categorisation of continuous variables (with imputed missing values). The final 16 categorical predictors included in the model were: age, gender, fever, new onset diarrhoea or vomiting, heart rate, respiratory rate, $FiO_2$, temperature, albumin, C-reactive protein, eGFR, pH, monocytes, WBC, frailty score, and Glasgow Coma Scale score.

AUROC was 0.893 (95% CI 0.864 to 0.922) (table 2). At a 20% predicted probability of ITU admission, sensitivity was 79% (95% CI 74 to 84), specificity was 83% (95% CI 81 to 84), positive predictive value was 49% (95% CI 46 to 52), and negative predictive was 95% (95% CI 94 to 96) (table 3B). Calibration was good; a calibration plot is shown in figure 1B, and the calibration slope was 0.91 (95% CI 0.80 to 1.01) (table 2). Model coefficients are presented in online supplemental table 5.

Addition of comorbidities to the predictors included in the model did not improve performance.

## Reduced UHB model and external validation in the CovidCollab dataset

A total of 6099 patients admitted with COVID-19 were included in the CovidCollab external validation dataset; 1668 (27%) died and 722 (12%) were admitted to ITU (table 1 and online supplemental table 1). Not all variables included in the UHB model derived previously were available in the CovidCollab dataset. Therefore, revised and reduced models were developed in UHB data using the subset of candidate predictors common to both the UHB and CovidCollab datasets (reduced UHB dataset, UHB-R), using logistic regression with stepwise selection, and these were then externally validated in the Covid-Collab dataset.

The reduced set of 27 candidate predictors included demographic characteristics: age and gender; symptoms: cough, fever and delirium; physiological measures and vital signs: BMI, systolic blood pressure, diastolic blood pressure, heart rate, temperature, respiratory rate, oxygen saturation, $FiO_2$ and chest X-ray; frailty score; Glasgow Coma Scale score; laboratory test results: eGFR, pH, base excess, lymphocytes, neutrophil:lymphocyte ratio, haemoglobin, bicarbonate, C reactive protein, alanine aminotransferase, urea and lactate.

## Mortality (28 days)

For the 28-day mortality outcome, following stepwise selection, the final 10 categorical predictors (common to both datasets) included in the reduced logistic regression model were age, oxygen saturation, $FiO_2$, respiratory rate, temperature, systolic blood pressure, C reactive protein, pH, urea and frailty score.

The selected predictors were a subset of those in the original UHB model derivation, but gave similar model performance. AUROC in the UHB-R dataset was 0.791 (95% CI 0.761 to 0.822), and AUROC in the CovidCollab external validation dataset was 0.767 (95% CI 0.754 to 0.780) (table 2). At a 20% predicted probability of

**Table 3B** Sensitivity, specificity, PPV and NPV for intensive therapy unit admission within 28 days after admission (University Hospitals Birmingham derivation dataset)

| Predicted probability (%) | TP | TN | FP | FN | Sensitivity (95% CI) | Specificity (95% CI) | PPV (95% CI) | NPV (95% CI) |
|---|---|---|---|---|---|---|---|---|
| 10 | 161 | 617 | 240 | 22 | 0.878 (0.851 to 0.905) | 0.720 (0.694 to 0.746) | 0.401 (0.373 to 0.430) | 0.965 (0.956 to 0.974) |
| 20 | 144 | 709 | 148 | 39 | 0.789 (0.743 to 0.835) | 0.827 (0.814 to 0.841) | 0.494 (0.464 to 0.523) | 0.948 (0.937 to 0.959) |
| 30 | 129 | 771 | 86 | 54 | 0.707 (0.662 to 0.752) | 0.900 (0.891 to 0.908) | 0.601 (0.575 to 0.626) | 0.935 (0.926 to 0.944) |
| 40 | 115 | 797 | 60 | 68 | 0.631 (0.597 to 0.664) | 0.930 (0.925 to 0.935) | 0.659 (0.634 to 0.684) | 0.922 (0.915 to 0.929) |
| 50 | 97 | 817 | 40 | 86 | 0.532 (0.502 to 0.563) | 0.954 (0.952 to 0.956) | 0.711 (0.695 to 0.727) | 0.905 (0.900 to 0.911) |
| 60 | 81 | 831 | 26 | 102 | 0.445 (0.429 to 0.461) | 0.970 (0.963 to 0.977) | 0.760 (0.722 to 0.798) | 0.891 (0.889 to 0.893) |
| 70 | 67 | 841 | 16 | 116 | 0.367 (0.345 to 0.390) | 0.982 (0.975 to 0.989) | 0.812 (0.758 to 0.867) | 0.879 (0.876 to 0.882) |
| 80 | 55 | 848 | 9 | 128 | 0.303 (0.276 to 0.330) | 0.989 (0.985 to 0.994) | 0.860 (0.806 to 0.915) | 0.869 (0.865 to 0.874) |
| 90 | 35 | 854 | 3 | 148 | 0.190 (0.141 to 0.240) | 0.997 (0.993 to 1) | 0.924 (0.848 to 1) | 0.852 (0.844 to 0.860) |

FN, false negatives; FP, false positives; NPV, negative predictive value; PPV, positive predictive value; TN, true negatives; TP, true positives.

mortality, in the UHB-R dataset, sensitivity was 86% (95% CI 85% to 88%); specificity was 54% (95% CI 51% to 57%); PPV was 42% (95% CI 40% to 43%); and NPV was 91% (95% CI 90% to 92%); in the CovidCollab dataset, sensitivity was 88% (95% CI 87% to 89%); specificity was 46% (95% CI 45% to 47%); PPV was 38% (95% CI 0.37% to 0.38%); and NPV was 91% (95% CI 91% to 92%) (table 4A).

Calibration was good for both derivation and external validation datasets; calibration plots are shown in figure 1C,D, and calibration slopes were 0.89 (95% CI 0.81 to 0.97) and 0.85 (95% CI 0.75 to 0.94) for the UHB-R and CovidCollab datasets, respectively (table 2). Model coefficients are presented in online supplemental table 6.

### ITU admission
For the ITU admission outcome, the final 11 categorical predictors (common to both datasets) included in the reduced model were age, gender, fever, respiratory rate, $FiO_2$, C reactive protein, eGFR, pH, neutrophil:lymphocyte ratio, frailty score and Glasgow Coma Scale score.

AUROC in the UHB-R dataset was 0.906 (95% CI 0.883 to 0.929), and in the CovidCollab dataset was 0.811 (95% CI 0.795 to 0.828) (table 2). At a 20% predicted probability of ITU admission, in the UHB-R dataset, sensitivity was 83% (95% CI 81% to 85%); specificity was 83% (95% CI 82% to 84%); PPV was 51% (95% CI 49% to 52%); and NPV was 96% (95% CI 95% to 96%); in the CovidCollab dataset, sensitivity was 64% (95% CI 62% to 67%); specificity was 80% (95% CI 79% to 82%); PPV was 30% (95% CI 29% to 32%); and NPV was 94% (95% CI 94% to 95%) (table 4B).

Calibration was good for both derivation and external validation datasets; calibration plots are shown in figure 1E,F; calibration slopes were 0.94 (95% CI 0.84 to 1.04) and 0.95 (95% CI 0.82 to 1.08) for the UHB-R and CovidCollab datasets, respectively (table 2). Model coefficients are presented in online supplemental table 7.

### External validation of the ISARIC 4C score in the UHB dataset
The AUROC for the recently published ISARIC 4C score in the UHB dataset was 0.753 (95% CI 0.720 to 0.785). The calibration slope was 0.99 (95% CI 0.85 to 1.12) (table 2 and online supplemental figure 1).

It was not possible to externally validate the ISARIC 4C score in the CovidCollab dataset, as information on many of the comorbidities required to calculate the ISARIC comorbidity score was not available in the dataset.

### Sensitivity analyses
Analyses exploring different ways of handling missing data are reported in online supplemental appendix 2 and online supplemental figures 2 and 3.

### Complete case analysis
Few patients in the dataset had complete data (n=224/1040, 22%); model performance in this patient subset was slightly poorer for the mortality outcome:

AUROC 0.696 (95% CI 0.597 to 0.795) for mortality and 0.892 (95% CI 0.844 to 0.940) for ITU admission. Including patients with missing variables, with missing values imputed, improved model performance for predicting mortality; allowing even a single missing/imputed variable improved AUROC for mortality to 0.760 (95% CI 0.708 to 0.812) (online supplemental table 8).

### Stratification by gender and age
When patients were stratified by gender, the reduced models predicting mortality and ITU still performed well: AUROCs for mortality were 0.775 (95% CI 0.726 to 0.823) for males and 0.755 (95% CI 0.706 to 0.804) for females, and those for ITU 0.897 (95% CI 0.856 to 0.937) for males and 0.873 (95% CI 0.833 to 0.913) for females (online supplemental table 9).

When patients were stratified by age, the models performed slightly better in patients aged >60 years (AUROC 0.778 (95% CI 0.722 to 0.834) and 0.897 (95% CI 0.864 to 0.930) for mortality and ITU admission, respectively) compared with those aged ≤60 years (AUROC 0.730 (95% CI 0.638 to 0.823) and 0.845 (95% CI 0.761 to 0.930) for mortality and ITU admission, respectively).

### Time series analysis
Online supplemental figure 4 shows variation in logistic regression coefficients for the candidate predictors from day of admission and up to 7 days later. The majority of coefficients remained relatively constant over time. However, several (not necessarily statistically significant) trends in the modification of effects over the week of admission on mortality were visible, such as a decrease over the week of the effect of obesity on mortality, elevated effect of eosinophils, and an increase over the week of the effect of elevated haemoglobin, elevated potassium and elevated oxygen saturation. Some of these might be depletion effects related to relatively high patient mortality in the first few days, for example, the apparent protective effect of obesity and high eosinophils.

### DISCUSSION
Using routinely collected data for more than a thousand patients admitted with COVID-19 at a large UK hospital trust, we have developed and externally validated prognostic models for mortality and ITU admission. The models showed good discrimination and calibration. The candidate predictors explored included a clinically informed, wider range of demographics, clinical observations, symptoms, comorbidities, biomarkers and radiological investigations than those included in the derivation of existing prognostic scores or models.

If integrated into hospital electronic medical records systems, the model algorithms will provide a predicted probability of mortality or ITU admission within 28 days of hospital admission for each patient based on their individual data at, or close to, the time of admission, which will support clinicians' decision making with regard to

**Table 4A** Sensitivity, specificity, PPV and NPV for mortality at 28 days after admission for the reduced model (UHB derivation dataset and CovidCollab external validation dataset, using predictors common to both datasets)

| Predicted probability | TP | TN | FP | FN | Sensitivity (95% CI) | Specificity (95% CI) | PPV (95% CI) | NPV (95% CI) |
|---|---|---|---|---|---|---|---|---|
| UHB-R derivation dataset (reduced model) | | | | | | | | |
| 10 | 278 | 249 | 503 | 10 | 0.964 (0.955 to 0.972) | 0.332 (0.315 to 0.348) | 0.356 (0.351 to 0.360) | 0.960 (0.952 to 0.968) |
| 20 | 248 | 406 | 346 | 40 | 0.862 (0.847 to 0.878) | 0.539 (0.510 to 0.569) | 0.418 (0.403 to 0.433) | 0.911 (0.902 to 0.920) |
| 30 | 202 | 536 | 216 | 86 | 0.703 (0.671 to 0.734) | 0.713 (0.698 to 0.727) | 0.484 (0.464 to 0.504) | 0.862 (0.849 to 0.876) |
| 40 | 153 | 635 | 117 | 135 | 0.532 (0.511 to 0.553) | 0.844 (0.831 to 0.858) | 0.567 (0.547 to 0.587) | 0.825 (0.819 to 0.831) |
| 50 | 104 | 686 | 66 | 184 | 0.360 (0.335 to 0.384) | 0.913 (0.905 to 0.920) | 0.612 (0.593 to 0.632) | 0.788 (0.782 to 0.794) |
| 60 | 66 | 723 | 29 | 222 | 0.228 (0.211 to 0.246) | 0.962 (0.953 to 0.970) | 0.696 (0.663 to 0.729) | 0.765 (0.762 to 0.768) |
| 70 | 35 | 744 | 8 | 253 | 0.122 (0.101 to 0.144) | 0.990 (0.985 to 0.994) | 0.819 (0.753 to 0.884) | 0.746 (0.742 to 0.751) |
| 80 | 14 | 750 | 2 | 274 | 0.050 (0.036 to 0.064) | 0.997 (0.996 to 0.998) | 0.868 (0.835 to 0.900) | 0.733 (0.730 to 0.735) |
| 90 | 3 | 752 | 0 | 285 | 0.010 (0 to 0.022) | 0.999 (0.998 to 1) | 0.900 (0.573 to 1) | 0.725 (0.723 to 0.727) |
| CovidCollab external validation dataset | | | | | | | | |
| 10 | 1614 | 1218 | 3213 | 54 | 0.967 (0.962 to 0.973) | 0.275 (0.260 to 0.290) | 0.334 (0.330 to 0.338) | 0.957 (0.951 to 0.963) |
| 20 | 1471 | 2028 | 2403 | 197 | 0.882 (0.874 to 0.889) | 0.458 (0.445 to 0.471) | 0.380 (0.375 to 0.384) | 0.911 (0.907 to 0.916) |
| 30 | 1289 | 2721 | 1710 | 379 | 0.773 (0.763 to 0.782) | 0.614 (0.602 to 0.626) | 0.430 (0.425 to 0.435) | 0.878 (0.874 to 0.881) |
| 40 | 1059 | 3264 | 1167 | 609 | 0.635 (0.605 to 0.665) | 0.737 (0.717 to 0.756) | 0.476 (0.469 to 0.483) | 0.843 (0.835 to 0.851) |
| 50 | 850 | 3698 | 733 | 818 | 0.510 (0.470 to 0.549) | 0.835 (0.817 to 0.852) | 0.537 (0.523 to 0.551) | 0.819 (0.810 to 0.828) |
| 60 | 599 | 4025 | 406 | 1069 | 0.359 (0.305 to 0.414) | 0.908 (0.890 to 0.927) | 0.596 (0.578 to 0.615) | 0.790 (0.779 to 0.801) |
| 70 | 380 | 4222 | 209 | 1288 | 0.228 (0.186 to 0.270) | 0.953 (0.940 to 0.965) | 0.646 (0.626 to 0.666) | 0.766 (0.759 to 0.774) |
| 80 | 174 | 4355 | 76 | 1494 | 0.104 (0.077 to 0.132) | 0.983 (0.978 to 0.987) | 0.697 (0.671 to 0.722) | 0.745 (0.740 to 0.750) |
| 90 | 37 | 4414 | 17 | 1631 | 0.022 (0.006 to 0.039) | 0.996 (0.995 to 0.998) | 0.676 (0.572 to 0.780) | 0.730 (0.727 to 0.733) |

FN, false negative; FP, false positive; NPV, negative predictive value; PPV, positive predictive value; TN, true negative; TP, true positive; UHB, University Hospitals Birmingham; UHB-R, University Hospitals Birmingham reduced model.

**Table 4B** Sensitivity, specificity, PPV and NPV for intensive therapy unit admission within 28 days after admission in the reduced model (University Hospitals Birmingham derivation dataset and CovidCollab external validation dataset, using predictors common to both datasets)

| Predicted probability | TP | TN | FP | FN | Sensitivity (95% CI) | Specificity (95% CI) | PPV (95% CI) | NPV (95% CI) |
|---|---|---|---|---|---|---|---|---|
| UHB-R derivation dataset (reduced model) | | | | | | | | |
| 10 | 165 | 590 | 267 | 18 | 0.904 (0.877 to 0.931) | 0.689 (0.671 to 0.707) | 0.383 (0.368 to 0.398) | 0.971 (0.963 to 0.979) |
| 20 | 152 | 709 | 148 | 31 | 0.831 (0.812 to 0.849) | 0.827 (0.816 to 0.839) | 0.507 (0.494 to 0.519) | 0.958 (0.954 to 0.962) |
| 30 | 132 | 765 | 92 | 51 | 0.723 (0.698 to 0.749) | 0.893 (0.882 to 0.904) | 0.591 (0.567 to 0.615) | 0.938 (0.933 to 0.943) |
| 40 | 112 | 803 | 54 | 71 | 0.613 (0.573 to 0.653) | 0.937 (0.928 to 0.946) | 0.675 (0.649 to 0.702) | 0.919 (0.912 to 0.926) |
| 50 | 92 | 829 | 28 | 91 | 0.505 (0.483 to 0.526) | 0.967 (0.956 to 0.979) | 0.768 (0.708 to 0.829) | 0.901 (0.898 to 0.905) |
| 60 | 77 | 841 | 16 | 106 | 0.419 (0.403 to 0.435) | 0.981 (0.974 to 0.988) | 0.828 (0.779 to 0.877) | 0.888 (0.886 to 0.890) |
| 70 | 67 | 847 | 10 | 116 | 0.365 (0.344 to 0.386) | 0.988 (0.983 to 0.994) | 0.870 (0.816 to 0.924) | 0.879 (0.876 to 0.883) |
| 80 | 46 | 851 | 6 | 137 | 0.251 (0.189 to 0.313) | 0.993 (0.989 to 0.997) | 0.889 (0.840 to 0.938) | 0.861 (0.852 to 0.871) |
| 90 | 20 | 855 | 2 | 163 | 0.108 (0.067 to 0.149) | 0.997 (0.996 to 0.999) | 0.898 (0.838 to 0.957) | 0.840 (0.833 to 0.846) |
| CovidCollab external validation dataset | | | | | | | | |
| 10 | 574 | 3623 | 1754 | 148 | 0.794 (0.764 to 0.825) | 0.674 (0.651 to 0.697) | 0.247 (0.238 to 0.255) | 0.961 (0.956 to 0.965) |
| 20 | 465 | 4312 | 1065 | 257 | 0.644 (0.621 to 0.667) | 0.802 (0.785 to 0.819) | 0.304 (0.288 to 0.321) | 0.944 (0.941 to 0.947) |
| 30 | 377 | 4655 | 722 | 345 | 0.523 (0.497 to 0.549) | 0.866 (0.847 to 0.885) | 0.344 (0.320 to 0.367) | 0.931 (0.929 to 0.933) |
| 40 | 315 | 4881 | 496 | 407 | 0.437 (0.381 to 0.492) | 0.908 (0.887 to 0.928) | 0.389 (0.361 to 0.417) | 0.923 (0.918 to 0.929) |
| 50 | 250 | 5045 | 332 | 472 | 0.346 (0.258 to 0.434) | 0.938 (0.920 to 0.956) | 0.430 (0.413 to 0.447) | 0.914 (0.905 to 0.923) |
| 60 | 178 | 5168 | 209 | 544 | 0.247 (0.175 to 0.318) | 0.961 (0.945 to 0.977) | 0.462 (0.420 to 0.503) | 0.905 (0.898 to 0.912) |
| 70 | 113 | 5256 | 121 | 609 | 0.157 (0.105 to 0.208) | 0.978 (0.967 to 0.988) | 0.486 (0.439 to 0.534) | 0.896 (0.891 to 0.901) |
| 80 | 57 | 5319 | 58 | 665 | 0.079 (0.044 to 0.114) | 0.989 (0.984 to 0.995) | 0.500 (0.410 to 0.590) | 0.889 (0.886 to 0.892) |
| 90 | 18 | 5357 | 20 | 704 | 0.025 (0.006 to 0.044) | 0.996 (0.993 to 1) | 0.485 (0.367 to 0.603) | 0.884 (0.882 to 0.886) |

FN, false negatives; FP, false positives; NPV, negative predictive value; PPV, positive predictive value; TN, true negatives; TP, true positives; UHB-R, University Hospitals Birmingham reduced model.

appropriate patient care pathways and triage. This information might also assist clinicians in explaining complex prognostic assessments and decisions to patients and their relatives, particularly at times when relatives are unable to see the patient and understand how unwell they are.

## Summary of results

The models developed using all 63 available candidate predictors from UHB performed well with an optimism-adjusted AUROC of 0.779 (95 % CI 0.744 to 0.813) for mortality within 28 days of admission and 0.893 (95% CI 0.864 to 0.922) for ITU admission.

Not all variables included in the UHB dataset are routinely collected at admission in other hospitals; therefore, reduced models using only variables common to both UHB and the CovidCollab external validation dataset were explored. Discrimination remained similar, with an AUROC of 0.791 (95% CI 0.761 to 0.822) for mortality and 0.906 (95% CI 0.883 to 0.929) for ITU admission in the UHB derivation dataset. These reduced models also performed well in the CovidCollab external validation dataset, with AUROCs of 0.767 (95% CI 0.754 to 0.780) and 0.811 (95% CI 0.795 to 0.828) for mortality and ITU admission, respectively. The models also performed well in gender-stratified and age-stratified patient subgroups.

Calibration of all models showed good agreement between observed and predicted probabilities, particularly at lower predicted probabilities in the range where the models would be of most clinical utility.

We found that addition of comorbidities to the model predictors did not improve overall model performance. This may be due to a correlation between presence of comorbidities and related physiological measurements and/or biomarkers which are already captured by the model.

## Comparison with existing literature

Two systematic reviews summarised the existing secondary care COVID-19 prognostic models or scores published until 31 May 2020.[1 17] The majority of the reported models, along with several more recent ones,[18] were derived in Chinese cohorts. Many of the models included in the reviews demonstrated high discriminatory performance; however, all pre-existing models when assessed using the PROBAST score were at high risk of bias. Furthermore, few models were externally validated in suitable cohorts. By deriving our model from routinely collected data, we were able to reduce the risk of bias in patient selection as well as predictor and outcome measurements. Additionally, in this study, we were able to externally validate models in a large global heterogeneous cohort.

More recently, the most notable secondary care prediction model advised for uptake in UK hospitals was derived from the ISARIC–WHO collaborating cohort and has been externally validated.[5 19] Both the full and reduced UHB-derived models for mortality had slightly better discrimination than the ISARIC 4C score in the UHB data (AUROC 0.753, 95% CI 0.720 to 0.785 for 4C). This compares with an AUROC of 0.767 (95% CI 0.760 to 0.773) for the 4C score reported in the original ISARIC validation cohort.[5] However, better performance may be expected for models evaluated in their development dataset compared with external datasets. The newly developed UHB model offers an advantage over the ISARIC 4C model in that it uses only routinely collected patient data recorded at admission and does not require additional assessment and recording of specific comorbidities (which are often not routinely fully recorded at the point of admission).

In our time series analysis, we did not find strong evidence for trends in predictor coefficients over the first 8 days of admission, particularly for variables included in the final models, suggesting that time-dependent effects due to effect modification or selection bias in the first week are small. Another recent model derived from patients with COVID-19 in a Hong Kong hospital adopted the use of time-dependent routinely collected predictors; the model in the Hong Kong study demonstrated high discrimination, with an AUROC of 0.91 when predicting severe COVID-19 outcomes.[20] However, this model is yet to be peer-reviewed and externally validated.

## Strengths and limitations

The UHB dataset represents one of the largest and most ethnically diverse patient cohorts within the UK. Additionally, as part of the early UHB response to the COVID-19 pandemic, the hospital trust ensured that, on admission, all patients underwent a wide range of investigations to support international research efforts examining prognostic markers. This allowed us to examine a wide range of possible predictors (63 candidate predictors after exclusions). Lastly, a strength of this study was the good performance, in terms of both discrimination and calibration, of the simplified, reduced model in an externally validated cohort (CovidCollab), indicating its suitability for wider use, including potentially in LMICs.

Despite the strengths, the findings must be considered in light of the study's limitations. Although we were able to use a derivation dataset from UHB with low levels of missing data, the overall sample size was relatively small compared with that of the ISARIC study and was limited to one UK geographical location. However, we were able to externally validate the model in a larger external cohort. A second limitation was that in the external validation cohort, we were unable to examine all of the predictors included in the original full UHB model due to only a reduced set of candidate predictors being available in CovidCollab. Nevertheless, the model performed well and the results suggest it may be applicable in a wide range of datasets where only a reduced set of predictor variables is available. It was not possible to carry out stratified analysis by ethnicity as, in the UHB dataset, too few patients were included in most of the strata; ethnicity data were not available in the CovidCollab dataset. Our definition of 28-day COVID-19 mortality aligns with the current technical guidance from Public Health England and

the definition used by the UK government in reporting COVID-19 mortality statistics[21 22]; however, we acknowledge that this may not capture all COVID-19-related deaths, and some other studies have used a longer period of follow-up.[23]

## CONCLUSION

In this paper, we have described the development and external validation of novel prognostic models which predict mortality and ITU admission within 28 days of admission for patients admitted to hospital with COVID-19. The simple, reduced models used only routinely collected data gathered at admission, showed good discrimination and calibration, performed at least as well as the existing ISARIC 4C score and performed well in a validation cohort. The models can be integrated into existing electronic medical records systems to calculate each individual patient's probability of death or ITU admission at the time of hospital admission. The models should be further validated to determine their applicability in other populations. In addition, implementation of the models and clinical utility should be evaluated.

**Author affiliations**

[1]Institute of Applied Health Research, University of Birmingham, Birmingham, UK
[2]NIHR Birmingham Biomedical Research Centre, University Hospitals Birmingham NHS Foundation Trust, Birmingham, UK
[3]Department of Diabetes, Gartnavel General Hospital, Glasgow, UK
[4]Mahidol Oxford Tropical Medicine Research Unit, University of Oxford, Oxford, UK
[5]Centre for Anaesthesia Critical Care & Pain Medicine, University College London Hospitals NHS Foundation Trust, London, UK
[6]Institute of Inflammation and Ageing, University of Birmingham, Birmingham, UK
[7]University Hospitals Birmingham NHS Foundation Trust, Birmingham, UK
[8]National Institute for Health Research Biomedical Research Centre, Moorfields Eye Hospital NHS Foundation Trust, London, UK
[9]Health Data Research UK, London, UK

**Acknowledgements** MJP and YT are supported by the NIHR Birmingham Biomedical Research Centre. The views expressed are those of the authors and not necessarily those of the NHS, NIHR or the Department of Health and Social Care. We also acknowledge the collaborators for the CovidCollab study: Sarah Richardson and Miles Witham, AGE Research Group, NIHR Newcastle Biomedical Research Centre, University of Newcastle and Newcastle-upon-Tyne Hospitals NHS Foundation Trust, UK; Omar A Abdelwahab, Elsayed M Awad, Ahmed Y Azzam, Ahmed Cordie, Ahmed O Elmehrath, Mostafa El-Shazly and Almontacer E B Masood, Alazher University Hospital, Egypt; Osama M A S Abdulhadi, Hazem Ahmed, Muhammed Elhadi, Ahmed KM Hadreiez, Ahmed A Momen, Mosab A A Shaban and Alkhadra Hospital, Libya; Giuseppe Cecere, Aldo Rocca, Antonio Cardarelli Hospital, Italy; Hossam Aldein S Abd Elazeem, Mohammed H Abdelhafez, Islam A Ahmed, Shrouk M Elghazaly, Helal F Hetta, Mohamed Eltaher AA Ibrahim, Soha M Mohamed, Aliae A R Mohamed Hussein, Mohamed M Moustafa, Mariam Albatoul Nageh, Mahmoud M Saad, Alshaimaa M Saad and Omar Zein Elabedeen, Assiut University Hospital, Egypt; Victoria Cox, Danielle Hunsley, Rebecca Ryall, Kathleen T Shakespeare, Thyn Thyn and Rachael Webb, Barnsley Hospital NHS Foundation Trust, UK; Deepthy Hari Madhavan and Nik Sanyal, Birmingham Heartlands Hospital, UK; Bryony Brown and Matthew Hale, Bradford Teaching Hospitals NHS Foundation Trust, UK; Marie Goujon, Cambridge University Hospitals NHS Foundation Trust, UK; Benjamin Jelley, Cardiff University, UK; Laxmi Babar, Tina Doll, Agnieszka Felska, Daniel N Guerero, Sandeep Karthikeyan, Anne Karunatilleke, Helena Lee, Emma Livesey, Amelia Roberts and Charlotte Roberts-Rhodes, City Hospital, Birmingham, UK; Teresa Perra and Alberto Porcu, Cliniche San Pietro, A.O.U. Sassari, Italy; Antonio Buondonno, Giuseppe Cecere and Aldo Rocca, Department of Medicine and Health Science 'V. Tiberio', University of Molise, Italy; Vesna Hogan and Iain Wilkinson, East

Surrey Hospital, UK; Ioannis Baloyiannis, Jiannis Hajiioannou, Konstantinos Perivoliotis and George Tzovaras, General University Hospital of Larissa, Greece; Anna Fleck and Aine McGovern, Glasgow Royal Infirmary, UK; Victoria Gaunt, Gloucestershire Hospitals NHS Foundation Trust, UK; Laura Babb, Emily Bailey, Jay Darley, Ioan M Draghita, Alexander Hickman, Jason Kalloo, Akhil Kanzaria, Katy Madden, Wasim Nawaz and Ambreen Sadiq, Good Hope Hospital, UK; Rifa Cardoso, Margherita Faulkner, William Hurst, Ellen James, Aimee Leadbetter, Jordan Mayer, Tanya Robinson, Emma Stratton, Miriam Thake, Hannah Thould and Hannah Watson, Great Western Hospital, UK; Ravindra Belgamwar and Corrina Bentley, Harplands Hospital, UK; Sarah Freshwater, Health Education West Midlands, UK; Sergio DV Ruiz, Nuria M Sanz, Milagros Carrasco Prats, Pedro V F Fernández, Clara G Francés, Esther M Manuel, Miguel R Marin, Pedro L Morales, Patricia P Pérez, María V Soriano and Ismael Mora-Guzmán, Hospital General Reina Sofía, Spain; Ismael Mora-Guzmán, Hospital Santa Bárbara, Spain; Melanie Dani, Imperial College Healthcare NHS Trust, UK; Fabio Barra, Antonella Ferraiolo, Simone Ferrero, Claudio Gustavino and Chiara Kratochwila, IRCCS Ospedale Policlinico San Martino, Italy; Eric W Etchill, Alodia Gabre-Kidan, Joshua H Gray, Elliott R Haut, Harsha Malapati, Sarah F Rapaport, Kent A Stevens and Dominique Vervoort, Johns Hopkins Hospital, USA; Mohammed A Azab, King Abdullah Medical City Specialist Hospital, Saudi Arabia; Catherine Bryant, Hannah Cheney-Lowe, Catrin Cox, Andrew Crowe, Gordon Dick, Sarah Evans, Patrick CP Hogan, Kar Yee Law, Alexandra Richardson, Fabio Speranza, Kathryn Toppley, Julie Whitney and Eirene Yeung, King's College Hospital, UK; Mary Ni Lochlainn and Claire Steves, King's College London, UK; Alexandros Charalabopoulos, Spyridon Davakis, Amalia Karapanou, Theodore Liakakos, Eustratia Mpaili, Maria Mpoura, Michail A Sampanis and Nikolaos V Sipsas, Laiko University Hospital, Greece; Lucy Beishon, Elinor Burn, Parveen Doddamani, Victoria Haunton, Shahriar Kabir, Hannah Shaw and Chloe Warner, Leicester Royal Infirmary, UK; Chee Soo, Maidstone and Tunbridge Wells NHS Trust, UK; Yasmin K NasrEldin, Minia University Hospital, Egypt; Nourhan A A Ghannam, Minya General Hospital, Egypt; Isobel Sleeman, NHS Grampian, UK; Ravindra Belgamwar and Corrina Bentley, North Staffordshire Combined Healthcare NHS Trust, UK; Ali Ali, Sylvia Amini, James Belcher, Marie Giles, Hayley Jarvis, Nathan Jenko, Suvira Madan, Alexander Noar, Favour Nwolu, Jessica Parkin, Lauren C Passby, Jarita Sivam, Michael Surtees, Joanne Wagland, Ruth West and David Williams, Northern General Hospital, UK; Avinash Aujayeb, Lindsey Dew, Catherine Dotchin, James M Dundas, Elinor Edwards, Georgia F Gilbert, Karl Jackson, Sarah H Manning, Dominic Maxfield, Nicholas Moss, Declan C Murphy, Ellen Tullo and Sarah H Welsh, Northumbria NHS Hospital Trust, UK; Tahir Masud, Nottingham University Hospitals NHS Trust, UK; Mustafa Alsahab, Oxford University Hospitals NHS Trust, UK; Antonio Buondonno and Enrico Pinotti, Policlinico San Pietro, Italy; Francesco Alessandri, Gioia Brachini, Giancarlo Ceccarelli, Flavia Ciccarone, Pierfranco M Cicerchia, Bruno Cirillo, Giorgio De Toma, Giulia Duranti, Enrico Fiori, Giovanni B Fonsi, Pierfrancesco Lapolla, Simona Meneghini, Andrea Mingoli, Francesco Pugliese, Paolo Sapienza, Luigi Simonelli and Martina Zambon, Policlinico Umberto I, Sapienza University of Rome, Italy; Caterina Cattel and Laurenny Guzman, Princess Royal Hospital, King's College Hospital Trust, Surrey, UK; Hannah Dowell, Aina Ibukunoluwakitan, Fawsiya Mohamed, Claire Spice and Amanda Stafford, Queen Alexandra Hospital, Portsmouth, UK; Jolene Atia, Catherine Atkin, Hannah Currie, Felicity Evison, Heena Khiroya, Zeinab Majid and Maria Qurashi, Queen Elizabeth Hospital Birmingham, UK; Siobhan Coulter, Claire McDonald, Georgina Muir, Catherine O'Mahony and Caroline Tait, Queen Elizabeth Hospital Gateshead, UK; Rowan Davies, Katie Honney and Laura Winter, Queen Elizabeth Hospital King's Lynn, UK; Olubayode Adewole, Queen's Hospital Romford, UK; Amir Abdelmalak, Mohammed Ahmad, Muhammed H Ansari, Kingsley Appiah, Rajesh Dwivedi, Hope Elrick, Hedra Ghobrial, Rosie Jackson, Sophie Jeffs, Sasha Jeyakumar, Eleanor Lunt, Bushra Muzammil, Sylvia Pytraczyk, Jonathan Sheldrake, Jennifer Smith, Hannah Tobiss, Mark Vettasseri, Ruth H Willott and Hein Zaw, Queens Medical Centre, Nottingham, UK; Katherine Patterson, Queen's University Belfast, UK; Moulinath Bannerjee, Jean Cummings, Barbara Hart and Tom Maughan, Royal Bolton Hospital, UK; Clare Baguneid, Gabrielle Budd, Lizzie Moriarty, Omoteniola Odutola and Hannah Street, Royal Derby Hospital, UK; Alexis Carr, Royal Devon and Exeter Hospital, UK; Jennifer Pigott, Royal Free London NHS Foundation Trust, UK; Sarah Baldwin, Hannah Bashir, Jake Gibbon, Amy Gray, Grace Lewis, Christina Page and Rosanna Varden, Royal Victoria Infirmary, UK; Anthony Grubb, Elizabeth Holmes, Harjinder Kainth, Natalie McNeela, Lara Reilly, Abigail Reynolds and Mark Whitsey, Royal Wolverhampton NHS Trust, UK; Mertcan Akcay, Yeşim Akdeniz, Emrah Akın, Fatih Altintoprak, Zülfü Bayhan, Recayi Capoglu, Hakan Demir, Necattin Firat, Emre Gonullu, Tarık Harmantepe, Baris Mantoglu, Ali Muhtaroglu, Merve Yigit and Yasin A Yildiz, Sakarya Faculty of Medicine, Turkey; Lobna Al-Sodani, Nicole Burden, Evelyn Charsley, Thomas Kneen, Angeline Price and Emma Swinnerton, Salford Royal Hospital, UK; Yen Nee J Bo, Hayley R Boden, Reem Bulla, Alison Eastaugh, Helena Lee, Asma Khan, Mohammed Mubin, Amelia Roberts, Anthony Umeadi, Stephanie

Wallis, Megan Williamson and Yu Lelt Win, Sandwell General Hospital, UK; Eltayeb A Ahmed, Abdulmoiz Aljafari, Abdulmalek Aljafari, Abdulkader Mohammad, Sharq Alneel Hospital, Sudan; Manpreet Badh, Amy Birchenough, Nick Coulthard, Alice Devaney, Ratnam Gandhi, Katharine Hood, Samuel North, Martha Pinkney, Ellie Shaw and Elisha Whelan, Solihull Hospital, UK; Adam Seed, Southport and Ormskirk Hospital NHS Trust, UK; Gurinder Dogra, Claire Morris and Rebecca Wright, South Tyneside District Hospital, UK; Stephen Lim, Lia Orlando, Harnish Patel, Prabhleen Puri and Sing Yang Sim, Southampton General Hospital, UK; Carolyn Akladious, Gitanjali Amaratungaz, Taha Amir, Cheran Anandarajah, Rachael Anders, Sally Aziz, Anna Barnard, Monica Bawor, Laura Bremner, Hannah Bridgwater, Hejab Butt, Andra Caracostea, Theodore Chevallier, Victoria Comerford, Jack Cullen, Niamh Cunningham, Daniel Curley, Madeleine Daly, Nikhita Dattani, Benyamin Deldar, Arjun Desai, Nirali Desai, Jugdeep Dhesi, Maria Dias, Hannah C Dooley, Samiullah Dost, Hiren Dusara, Alexander Emery, Cassandra Fairhead, Antia Fernandez, Gracie Fisk, Madeleine Garner, Hannah Gerretsen, Andrew Ghobrial, Zaynub Ghufoor, Deirdre Green, Charlotte Greene, Karla Griffith, Ayushi Gupta, Patrick Harrison, Aidan Haslam, Torben Heinsohn, Lindsay Hennah, Abigail Hobill, Katherine Hopkinson, Lara Howells, Nicole Hrouda, Irem Ishlek, Rishi Iyer, Nuha Kardaman, Mairead Kelly, Nicola I Kelly, Hesham Khalid, Muhammad S Khan, Haris Khan, Matthew King, Li Kok, Aneliya Kuzeva, Rebecca Lau, Gabriel Lee, Gavriella Levinson, Danielle Lis, Baguiasri Mandane, Jamie Mawhinney, Henry Maynard, Sophie Mclachlan, Michelle Metcalf, John Millwood-Hargrave, Kelvin Miu, Aaliya Mohammed, Hamilton Morrin, Stephanie Mulhern, Daniel Muller, Varun Nadkarni, Hanna Nguyen, Alice O'Docherty, Sinead O'Dwyer, Marc Osterdahl, Ismini Panayotidis, Shefali Patel, Rose Penfold, Rupini Perinpanathan, Dina Radenkovic, Thurkka Rajeswaran, Tahmina Razzak, Emily Ross-Skinner, Hazel Sanghvi, Ross Sayers, Luca Scott, Sri Sivarajan, Katharine Stambollouian, Jack Stewart, Amybel Taylor, Hrisheekesh Vaidya, Vittoria Vergani, Madiha Virk, Vaishali Vyas, Eleanor Watkins, Catherine Wilcock, Mettha Wimalasundera, Stephanie Worrall, Natalie Yeo and Humza Yusuf, St Thomas' Hospital, UK; Adam H Dyer, Cliona Ni Cheallaigh and Liam Townsend, St. James's Hospital, Ireland; Jocelyn Amer, Emily Lyon and Michael Sen, Sunderland Royal Hospital, UK; Mohammed Al-Sadawi, Adam Budzikoski, Ishmam Ibtida and Yusra Qaiser, SUNY Downstate Brooklyn, USA; Mohammad T Azam, Asad J Choudhry and William Marx, SUNY Upstate University Hospital, USA; Ahmad Bouhuwaish and Ahmed SA Taher, Tobruk Medical Center, Libya; Nikolaos Georgiou, Jade Man, Paul Reynolds and Benjaman To, Tunbridge Wells Hospital, UK; Fatma D Collins, Sharon Budd, Ellanna Griffin, Yue Guan, Deevia Hanji, Lily Lowes, Awolkhier Mohammedseid-Nurhussien, Farhana Moomo, Olebu Ogochukwu and Katie Thin, University Hospitals Coventry and Warwickshire NHS Trust, UK; Elinor Burn, University Hospitals of Leicester NHS Trust; Terry Hughes, Thomas A Jackson, Laura Magill, Lauren McCluskey, Hannah Moorey, Kelvin Okoth, Rita Perry, Michala Petitt, Thomas Pinkney and Daisy Wilson, University of Birmingham; Grace ME Pearson, University of Bristol, UK; Christopher N Osuafor and Kelli Torsney, University of Cambridge, UK; David Strain, Jane Masoli, University of Exeter, UK; Jenni Burton and Terence Quinn, University of Glasgow, UK; Lucy Beishon, University of Leicester, UK; Joanne Taylor, University of Manchester, UK; Adam Gordon, University of Nottingham, UK; Gilda De Paola, Gaetano Gallo, Giuseppe Sammarco and Giuseppina Vescio, University 'Magna Graecia' of Catanzaro, Italy; Shivam Pancholi, University of Nicosia, Cyprus; Natalie Cox, University of Southampton, UK; Rajni Lal, Western Sydney Local Health District, Australia; Rand A Hussein, Zafaraniyah General Hospital, Iraq.

**Contributors** The study was designed by NJA, KN, ES, MJP, YT, CS and TT. Data were collected by KG, ES, CS and SG. TT, NJA and DG analysed the data. The first draft was written by NJA and TT. All authors critically reviewed and revised the manuscript. NJA acts as guarantor; the guarantor accepts full responsibility for the finished work and/or the conduct of the study, had access to the data, and controlled the decision to publish.

**Funding** This work was funded by the Medical Research Council UK Research and Innovation (reference COV0306). The funder had no role in developing the research question or the study protocol.

**Competing interests** NA, ES, KN, MJP, AKD, CS, TT and YT report a grant from UKRI MRC during the conduct of the study. ES reports grants from National Institute for Health Research (NIHR), Wellcome Trust, MRC, Health Data Research UK (HDR-UK), British Lung Foundation and Alpha 1 Foundation outside the submitted work. KN reports grants from MRC and HDR-UK outside the submitted work. DP reports grants from NIHR, MRC and Chernakovsky Foundation outside the submitted work.

**Patient consent for publication** Not applicable.

**Ethics approval** Ethical approval was provided by the East Midlands–Derby REC (reference: 20/EM/0158) for the PIONEER Research Database (data from University Hospitals Birmingham). For CovidCollab data, local, regional and national

approvals were obtained from all participating sites. In the UK, this study was registered as clinical audit or service evaluation, with approval granted in line with local information governance policies, in line with assessment and guidance by the Health Research Authority. At the lead site (University Hospitals Birmingham NHS Trust), this study was registered as clinical audit (CARMS-15986). In other countries, local principal investigators were responsible for obtaining approvals in line with their local, regional and national guidelines and recommendations. Only routinely collected data were collected and patient care was not altered by this study. Anonymised data were securely transferred to the Birmingham Centre for Prospective and Observational Studies, University of Birmingham via REDCap. All sites were required to confirm that approvals were in place prior to being provided with logins; written data sharing agreements were arranged where requested by individual sites.

**Provenance and peer review** Not commissioned; externally peer reviewed.

**Data availability statement** No data are available.

**ORCID iDs**
Nicola J Adderley http://orcid.org/0000-0003-0543-3254
Joht Singh Chandan http://orcid.org/0000-0002-9561-5141
Dhruv Parekh http://orcid.org/0000-0002-1508-8362
Elizabeth Sapey http://orcid.org/0000-0003-3454-5482

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
