## [Reviewer comments · BMJ Open]

ARTICLE DETAILS

TITLE (PROVISIONAL)	Development and external validation of prognostic models for COVID-19 to support risk stratification in secondary care
AUTHORS	Adderley, Nicola; Taverner, Thomas; Price, Malcolm; Sainsbury, Christopher; Greenwood, David; Chandan, Joht; Takwoingi, Yemisi; Haniffa, Rashan; Hosier, Isaac; Welch, Carly; Parekh, Dhruv; Gallier, Suzy; Gokhale, Krishna; Denniston, Alastair; Sapey, Elizabeth; Nirantharakumar, Krishnarajah

VERSION 1 – REVIEW

REVIEWER	De Matteis, Giuseppe Gemelli IRCCS
REVIEW RETURNED	18-Mar-2021

GENERAL COMMENTS	I thank the editor to be given the opportunity to revise this paper. I read with interest this manuscript regarding the development of a new score system for COVID-19 patients. The paper is well written, and the topic is timely and relevant considering the still ongoing pandemic. Moreover, due to the huge number of patients with COVID-19 admitted to our Emergency Departments, and the lack of available resources, the use of scoring systems to predict patient's deterioration is appreciated. However, there are some issues that should be addressed: • Introduction is too long and redundant and it could be reduced to increase readability. Moreover, most of the authors' statements are not supported by any references. Indeed, since research on Covid-19 is in continuous expansion, enhance the references could help this research both to fit better within the ongoing debate and to contextualize the authors statements.• Methods section is very hard to read. It should be structured by paragraph also to permit the readers understand many overlooked points:[ ] How patients were selected and enrolled? Do authors enrolled consecutive patients?[ ] How the time first onset of symptoms to hospital admission was recorded?[ ] Do authors know how many patients deceased after 28 days?[ ] Candidate predictor variables: here the study appear to be a little obscure. How were exactly chosen the variable in the model? If authors selected variables from prior experiences it should be specified in references. Moreover, since the study variables were very many, did authors evaluated well validated cumulative score
--

or other COVID-19 related score (comprising many variables in a single predictor?)

Model development: Maybe some parts could be included in supplementary materials. Moreover, although authors deeply describe model development, it is not clear how the results could be imputed in the model and how the results were showed. I understand from tables that model actually predicts overall risk from 0 to 100%. It should be better explained in this section.

Model discrimination and calibration. Although discrimination is well evaluated by ROC, calibration is much more difficult to evaluate in this setting. It could be more useful for clinician to have a reduced number of groups to evaluate (such as low risk, medium risk and high risk). In fact, obtaining such a categorization could help to better assess calibration, resulting also more helpful in a "real life" setting.

Sample size. A sample size analysis should be performed. Although the objective of the study is very ambitious, and the variables entered in the model as high as 63, the sample size is limited. Authors should elaborate.

• Results. Results section is unclear, and tables are very difficult to read. This part should be revised to improve readability even for clinicians with reduced statistical competences.

• Discussion. The discussion is very reduced and do not help to understand the value of the remarkable work behind this paper.

In my opinion authors should:

State in the beginning the specific value added by this paper to current research on Covid-19.

Compare the results of present paper with previous paper on the same argument, explaining why the proposed score should be better than previously proposed scores.

Explain the precise formula to calculate the score from selected variables.

State that the model prediction could be really useful especially in low resources settings/countries. I personally agree with this point, however if the calculation is very difficult or it need a complicate computerized system, it could be more useful a handy tool as WHO clinical scale.

The relevance of comorbidities in Covid-19 prognosis is well assessed. However, in this study the adding value of comorbidities did not affect the overall predicting values of the statistical model.

This should be discussed by the authors.

Moreover, about three out of four patients of the study cohort were older than 60 years, and almost all the deceased were older adults. The authors should discuss the score evaluation in this older cohort.

• References. Although many research was made on risk prediction in Covid-19, references section is very poor. At least some of these articles should be cited, compared and discussed:

Covino M, Sandroni C, Santoro M, Sabia L, Simeoni B, Bocci MG, Ojetti V, Candelli M, Antonelli M, Gasbarrini A, Franceschi F. Predicting intensive care unit admission and death for COVID-19 patients in the emergency department using early warning scores. *Resuscitation*. 2020 Nov;156:84-91. doi:

	10.1016/j.resuscitation.2020.08.124. Epub 2020 Sep 9. PMID: 32918985; PMCID: PMC7480278. □ Liang W, Liang H, Ou L, et al. Development and Validation of a Clinical Risk Score to Predict the Occurrence of Critical Illness in Hospitalized Patients With COVID-19. JAMA Intern Med 2020; 180(8):1081-1089. □ Haimovich A, Ravindra NG, Stoytchev S, et al. Development and validation of the quick COVID-19 severity index (qCSI): a prognostic tool for early clinical decompensation. Ann Emerg Med 2020. Doi: 10.1016/j.annemergmed.2020.07.022 [Epub ahead of print] □ Covino M, De Matteis G, Burzo ML, Russo A, Forte E, Carnicelli A, Piccioni A, Simeoni B, Gasbarrini A, Franceschi F, Sandroni C; GEMELLI AGAINST COVID-19 Group. Predicting In-Hospital Mortality in COVID-19 Older Patients with Specifically Developed Scores. J Am Geriatr Soc. 2020 Nov 16. doi: 10.1111/jgs.16956. Epub ahead of print. PMID: 33197278. □ Aliberti MJR, Covinsky KE, Garcez FB, Smith AK, Curiati PK, Lee SJ, Dias MB, Melo VJD, Rego-Júnior OFD, Richinho VP, Jacob-Filho W, Avelino-Silva TJ. A fuller picture of COVID-19 prognosis: the added value of vulnerability measures to predict mortality in hospitalised older adults. Age Ageing. 2021 Jan 8;50(1):32-39. doi: 10.1093/ageing/afaa240. PMID: 33068099; PMCID: PMC7665299.
--	--

REVIEWER	Harrison, Ewen University of Edinburgh
REVIEW RETURNED	25-Mar-2021

GENERAL COMMENTS	Many thanks for the opportunity to review this paper describing the development and validation of COVID-19 prognostic models. I do so from the point of view of having developed the ISARIC 4C mortality models. I like the paper and the authors have done a great job exploring the data. I would like to see this published, but I have major concerns about the models that have been generated as described below. 1) Purpose. a) The authors don't define the purpose of their models. Are these to be used in clinical practice for decision support? Or are these a research tool to stratify cohorts of individuals? This is important as it defines the modelling strategy. At the end of the discussion, the authors discuss the integration of models in an electronic patient record, which suggests they are for clinical use. Yet the motivation is stated as: "existing UK prognostic models for patients admitted to hospital with COVID-19 are limited by reliance on comorbidities, which are under-recorded in secondary care, and lack of imaging data among the candidate predictors." Why would healthcare staff looking after patients not have access to patient comorbidities? Given the comparison with the ISARIC scores, it should be pointed out that the ISARIC scores are not built using administrative data, but with prospectively collected data using a pre-specified protocol. It is likely therefore that comorbidities are better captured using these methods. It also seems odd to highlight the lack of imaging data, given that the final models presented here don't use imaging variables, so again this does not seem to be a particularly relevant basis for a new score. Is this a score to be used in clinical practice by healthcare staff? If it is for use with administrative datasets, then that should be clearly stated.
--

b) Frailty: just to extend this a bit. Frailty is exchangeable with comorbidities in our covid models. An issue with frailty vs comorbidity is that frailty is not validated in younger age groups, which is why we chose not to include it in 4C. Are the authors proposing that frailty be assessed in younger patients so that the score can be used clinically? Is this valid? The capture of frailty scores in administrative datasets is worse than comorbidities, so wouldn't be a good choice from that point of view.

2) Models.

a) The final models as intended for use are presented in the supplementary tables. I'm struggling with them. Using supplementary table 3 as an example. Age is included as a categorical variable, but the reference level (<30 year) has no events (no one dies). This almost certainly means the model will not have converged, as reflected in the odds ratios for other levels of age (e.g. odds ratio for death of 30-39 y compared with 30 y, OR = 735,275). Confidence intervals are not given (they should be), but likely tend to infinity on the upper bound. Why include a model that has been carefully honed through different approaches that appears to have fundamental mathematical issues, especially when it is easily solved by collapsing the lower levels of the factor? Perhaps I am missing something and apologise if I am.

b) There are clear issues with the categorisation of some of the continuous variables. For instance, for oxygen saturations, patients with levels 80-88% are at greater risk of death than those with saturations <80%. This is not reflected in the continuous GAMs provided in the supplement. Similarly, patients with a pH<7.30-7.34 (acidosis) are at lower risk of death than those with a pH in the normal range (there is wiggle in the GAM, although this is uninterpretable given the inclusion of lactate, base excess, and bicarbonate in the same model). Have these models been sense-checked by a clinician? What is the explanation for patients with acidosis or profound hypoxia having a lower risk of death than those who are not? There may be some reason, but I think it more likely that this represents overfitting and these should not have been incorporated into final models. A model describing worse physiological parameters being associated with better survival will have patient safety implications.

c) I can't see a table with the ICU admission outcome stratified by other variables (which should be included), but there is presumed to be a similar no/low event rate issue with >90 y.

3) Imputation. It is not clear how imputation was done, I couldn't see the variables that contributed to the models described. Were outcomes included, for instance? Predictive mean matching is described as the method, which would be the case for the continuous missing data, but presumably not for categorical missing data? Most importantly, reference is made to Rubin's rules for combining models, but the table footnotes report "coefficients derived after a single imputation of missing data"? Does that mean the presented final models are not derived using multiple imputation as is described? This is clearly problematic and represents a high risk of bias.

4) Comparison of models: the external validation of the 4C mortality score is greatly appreciated, many thanks. By definition, a new score derived in a particular dataset will perform better than an external score. What would be useful, is to compare both models in an external dataset. Then the added discrimination provided by the

	extra variables included can be clearly determined. It is unfortunate that a comparison in external validation cohort is not possible, and therefore conclusions around "out performance" need to be tempered by the limitation in methods. 5) XGboost models: the authors make a number of statements around validation and recalibration of the ISARIC XGboost model in their data. I'm a little perplexed about this, as the models are not published. It may be that the authors have used the same variables in their own dataset to derive a gradient boosted tree model and that is what is reported. This though has nothing to do with our own models, as none of the weights are used. All references to this would be best removed. 6) Calibration: The 4C mortality model is simple by design and can be improved upon at the expense of added complexity. An important question we and others are looking at is performance in patient subgroups poorly represented in ISARIC, both in terms of discrimination and calibration. It would be great therefore to see an analysis capturing low discrimination and calibration using 4C in, say, different age groups, sex, socioeconomic groups, ethnicities etc. That would help clinicians identify subgroups not well described by the model. This has not been done and it would be great if it could be. Calibration is most useful in the external dataset and including subgroups. It is almost pointless to test calibration in the training dataset. Calibration in the large (intercept) (not reported) and slope should both be reported. The calibration plot in the supplement didn't have a legend in my version so I couldn't quite work out what was what. It wasn't clear how the imputed datasets had been treated in the calibration analysis. This is obviously crucial given the variation that can be seen in calibration across imputed sets. 7) Time series. From what I understand, multiple logistic regressions models have been created over time and the coefficients for particular variables compared. I like the idea, but it is the stuff of causal inference nightmares. The reader is invited to draw conclusions about the effect of a particular characteristics on the probability of death on a given day, but this is as likely to reflect the changing population over time (patients die and are excluded) as anything meaningful about the contribution of particular characteristics. I think this would be best removed. 8) 4C deterioration score: Can the authors compare the ITU score with the 4C deterioration score? https://www.thelancet.com/pdfs/journals/lanres/PIIS2213-2600(20)30559-2.pdf In summary, therefore, I greatly appreciate the work the authors have done here. The final models appear to have significant issues and are not usable. I would be glad to look at a fully revised approach to this. Many thanks again.
--	---

VERSION 1 – AUTHOR RESPONSE

Reviewer 1:

I thank the editor to be given the opportunity to revise this paper.

I read with interest this manuscript regarding the development of a new score system for COVID-19 patients. The paper is well written, and the topic is timely and relevant considering the still ongoing pandemic. Moreover, due to the huge number of patients with COVID-19 admitted to our Emergency Departments, and the lack of available resources, the use of scoring systems to predict patient's deterioration is appreciated. We thank the reviewer for the positive feedback.

However, there are some issues that should be addressed:

Introduction is too long and redundant and it could be reduced to increase readability. Moreover, most of the authors' statements are not supported by any references. Indeed, since research on Covid-19 is in continuous expansion, enhance the references could help this research both to fit better within the ongoing debate and to contextualize the authors statements.

We have now edited the introduction, removing unnecessary information, clarifying some sentences, and adding references to additional recent studies. We thank the reviewer for the

2

suggested references, several of which are now mentioned in either the Background or Discussion.

Methods section is very hard to read. It should be structured by paragraph also to permit the readers understand many overlooked points:

We have now revised and clarified the Methods in line with your comments, as outlined below, including addition of subheadings to help the reader navigate more easily.

- How patients were selected and enrolled? Do authors enrolled consecutive patients?

Anonymised, routinely collected data for all patients with COVID-19 admitted to UHB was included. For CovidCollab, data collection was dependent on the specific processes within individual participating hospitals and the capacity of the data collector. This information has now been added in the Methods.

- How the time first onset of symptoms to hospital admission was recorded?

Information on symptoms was collected at the point of admission; this is noted in the Data source section of the Methods.

- Do authors know how many patients deceased after 28 days?

The current definition of COVID-19 mortality used by the UK government to report COVID19 mortality statistics – a definition recommended within technical guidance from Public Health England – is death within 28 days of the first positive COVID-19 test. In our datasets, patients underwent COVID-19 tests at the point of admission, and therefore our outcome definition was mortality within 28 days after admission. Only patients with at least 28 days' follow-up were included in our analysis. Some patients did die after 28 days, but this may have been unrelated to COVID-19. Furthermore, this information is only available for patients admitted earlier in the study period, as we do not have longer follow-up for the

whole dataset; therefore any data we can provide would be limited to a subset of included patients. Among the 288 patients in the UHB dataset who died within 28 days, the median (IQR) time to death was 8.1 (4.0-17.7) days.

We have now added the following to the Strengths and limitations section: 'Our definition of 28-day COVID-19 mortality aligns with the current technical guidance from Public Health England and the definition used by the UK government in reporting COVID-19 mortality statistics;^{1,2} however, we acknowledge that this may not capture all COVID-19-related deaths, and some other studies have utilised a longer period of follow-up.³

- Candidate predictor variables: here the study appear to be a little obscure. How were exactly chosen the variable in the model? If authors selected variables from prior experiences it should be specified in references. Moreover, since the study variables were very many, did authors evaluated well validated cumulative score or other COVID-19 related score (comprising many variables in a single predictor?)

3

As noted in the Candidate predictor variables section, the initial list of candidate predictors was compiled with reference to previous literature (primarily the previously developed prognostic models cited in the Background), discussion with clinical experts (specialists in acute care, critical care and geriatric medicine), and based on availability of variables routinely collected in secondary care/at UHB.

We did not combine multiple variables into a single predictor, as, after consultation with clinical experts, the team agreed that more granular-level information would have greater clinical applicability.

- Model development: Maybe some parts could be included in supplementary materials. Moreover, although authors deeply describe model development, it is not clear how the results could be imputed in the model and how the results were showed. I understand from tables that model actually predicts overall risk from 0 to 100%. It should be better explained in this section.

Thank you for your recommendation. We have now moved the paragraph on handling of continuous variables with non-linear associations to Supplementary Appendix 1. We have also expanded the description of the multiple imputation method used (Methods, Missing data section) as follows: 'missing continuous variables (vital signs, laboratory tests) and symptoms were imputed using multiple imputation using chained equations (using the R "mice" multiple imputation package). We performed 5 imputations and a maximum of 50 iterations.⁴

Continuous variables were imputed with predictive mean matching, and categorical variables with logistic regression (logreg) or polytomous regression (polyreg). Input variables for the multiple imputation included all available candidate predictor variables in the dataset; outcomes were not included in the imputation variables.'

We have now also made it explicit in the aims (in the final paragraph of the Background, Aims and rationale) that the derived models calculate predicted probability of the outcomes at an individual patient level: 'Specific objectives were: (i) to develop novel prognostic models for calculating predicted probability of adverse outcomes (death, intensive therapy unit (ITU) admission) at an individual patient level in a UK secondary care setting'.

- Model discrimination and calibration. Although discrimination is well evaluated by ROC, calibration is much more difficult to evaluate in this setting. It could be more useful for clinician to have a reduced number of groups to evaluate (such as low risk, medium risk and high risk). In fact, obtaining such a categorization could help to better assess calibration, resulting also more helpful in a “real life” setting.

The calibration plots compare the predicted probability with the observed probability of the outcome at different probability quantiles. Sensitivity and specificity at different predicted probability deciles are also presented in Table 3.

During the development of the models, we discussed clinical utility with a number of secondary care clinicians, and they consistently expressed a preference for being provided with the absolute predicted probability of the outcome (death or ITU admission) for an
4

individual patient rather than a summary risk level (such as high, medium or low risk). We therefore chose not to reduce the output to this type of categorisation, but rather provide the more detailed level of information to support clinical judgement.

- Sample size. A sample size analysis should be performed. Although the objective of the study is very ambitious, and the variables entered in the model as high as 63, the sample size is limited. Authors should elaborate.

We did not perform a sample size calculation prior to the study, as we used all available data up to the start of our study. However, we have now used Riley et al's (BMJ 2020;368:m441) sample size calculation methods for logistic regression models, and assuming a 25% binary outcome rate (for the primary mortality outcome) with 27 candidate predictors (the number of candidate predictors in the reduced, externally validated models) and a conservative R² CS of

0.5, the required sample size is 562. Our derivation sample size was 1040.

Results. Results section is unclear, and tables are very difficult to read. This part should be revised to improve readability even for clinicians with reduced statistical competences.

We have now edited the results section to try to improve readability by reducing paragraph length, moving some information to Supplementary Appendix 2, and introducing subheadings. We believe it is important to retain the various measures of model performance to allow the reader to fully appraise the models; however we hope the restructuring now makes it easier for clinicians to find the parts most relevant to them.

Discussion. The discussion is very reduced and do not help to understand the value of the remarkable work behind this paper.

We thank the reviewer for their kind words. To address this point we have now edited the discussion to make it clearer and more readable, by introducing subheadings, removing unnecessary sentences, and clarifying several points.

In my opinion authors should:

- State in the beginning the specific value added by this paper to current research on Covid-19.

Thank you. We have now clarified this under a new Aims and rationale subheading at the end of the Background; this reads as follows:

'To date there have been few prognostic models for patients admitted to hospital with COVID-19 developed in a UK dataset. Furthermore, evaluation of the extent to which the inclusion of comorbidities, imaging and additional biomarkers improves model performance is required. It also remains to be determined whether updating the clinical parameters with evolving biomarkers improves prediction of the clinical course of patients as the disease evolves.

'The overarching aim of this study was to develop prognostic models for patients admitted to hospital with COVID-19 using routinely collected data at the point of admission, which can

5
be used in a secondary care setting to support clinical decision-making. Specific objectives were: (i) to develop novel prognostic models for calculating predicted probability of adverse outcomes (death, intensive therapy unit (ITU) admission) at an individual patient level in a UK secondary care setting; (ii) to externally validate these models in an international dataset (including data from UK hospitals); (iii) to externally validate the existing UK ISARIC 4C score;

and (iv) to compare performance of the newly developed models with the UK ISARIC 4C score. In addition, we developed daily models using time series data from the first eight days from admission to explore changes in predictors over time.'

- Compare the results of present paper with previous paper on the same argument, explaining why the proposed score should be better than previously proposed scores.

The only previous UK study to date evaluating similar outcomes in hospitalised patients with COVID-19 is the ISARIC study. Comparison with the findings of the ISARIC study, together with evaluation of the ISARIC 4C score in our dataset, is integral to our paper and the study is mentioned and commented on numerous times. Many other studies, largely in non-UK populations, are also cited and discussed in the Background and Discussion sections. We have also added mention of the studies the reviewer kindly drew our attention to (see below).

- Explain the precise formula to calculate the score from selected variables.

We have now added the logistic regression formula above Supplementary Table 3, together with the formula for calculating probability. Coefficients for the equations for all prognostic models are provided in Supplementary Tables 3 and 5-7.

- State that the model prediction could be really useful especially in low resources settings/countries. I personally agree with this point, however if the calculation is very difficult or it need a complicate computerized system, it could be more useful a handy tool as WHO clinical scale.

We agree that the models may be useful in low resource countries, but we have refrained from overemphasising this as we believe further validation in low and middle-income countries (LMIC) is required first. While the external validation dataset (CovidCollab) does include some data from LMICs, it also includes data from higher income countries.

The relevance of comorbidities in Covid-19 prognosis is well assessed. However, in this study the adding value of comorbidities did not affect the overall predicting values of the statistical model. This should be discussed by the authors.

Thank you. We have now added the following to the Discussion: 'We found that addition of comorbidities to the model predictors did not improve overall model performance. This may be due to a correlation between presence of comorbidities and related physiological measurements and/or biomarkers which are already captured by the model.'

Moreover, about three out of four patients of the study cohort were older than 60 years, and almost all the deceased were older adults. The authors should discuss the score evaluation in this older cohort.

6

Thank you for your suggestion. We have now evaluated the performance of the model in subgroups of patients aged ≤ 60 years and >60 years. The model performed well in both, but slightly better in the >60 year age group (although confidence intervals overlapped). This is now presented in the Sensitivity analyses section of the Results and in Supplementary Table 9.

References. Although many research was made on risk prediction in Covid-19, references section is very poor. At least some of these articles should be cited, compared and discussed

Thank you for your recommendations. The majority of these studies were published while we were in the external validation and write-up stage of our analysis, and so they did not inform our predictor selection. However, we have now mentioned Covino et al (Resuscitation 2020), along with another study on NEWS2, in the Background, and referenced Liang et al (JAMA Int Med 2020), Covino et al (J Am Geriatr Soc 2020) and Aliberti et al (Age Ageing 2021) in the Discussion. Haimovich et al explored a different outcome (respiratory failure) in a subset of patients with minimal oxygen requirements, and has not been cited.

Reviewer 2:

Many thanks for the opportunity to review this paper describing the development and validation of COVID-19 prognostic models. I do so from the point of view of having developed the ISARIC 4C mortality models. I like the paper and the authors have done a great job exploring the data. I would like to see this published, but I have major concerns about the models that have been generated as described below. We thank the reviewer for the positive feedback and for the insightful comments below. We hope we have responded to these satisfactorily.

1) Purpose.

a) The authors don't define the purpose of their models. Are these to be used in clinical practice for decision support? Or are these a research tool to stratify cohorts of individuals? This is important as it defines the modelling strategy. At the end of the discussion, the authors discuss the integration of models in an electronic patient record, which suggests they are for clinical use. Yet the motivation is stated as: "existing UK prognostic models for patients admitted to hospital with COVID-19 are limited by reliance on comorbidities, which are under-recorded in secondary care, and lack of imaging data among the candidate predictors." Why would healthcare staff looking after patients not have access to patient comorbidities? Given the comparison with the ISARIC scores, it should be pointed out that the ISARIC scores are not built using administrative data, but with prospectively collected data using a pre-specified protocol. It is likely therefore that comorbidities are better captured using these methods. It also seems odd to highlight the lack of imaging data, given that the final models presented here don't use imaging variables, so again this does not seem to be a particularly relevant basis for a new score. Is this a score to be used

7

in clinical practice by healthcare staff? If it is for use with administrative datasets, then that should be clearly stated.

Thank you. We have now clarified the intended purpose of the models in a revised Aims and rationale section, at the end of the Background, as follows: 'The overarching aim of this study was to develop prognostic models for patients admitted to hospital with COVID-19 using routinely collected data at the point of admission, which can be used in a secondary care setting to support clinical decision-making.'

We acknowledge that secondary care clinicians have access to comorbidity information, and that many comorbidities are often documented in the discharge codes. However, much of this information is gathered and recorded during the patient stay; at the point of patient admission, this information may not be available in the routinely collected and recorded patient data – and our aim was to use routine data available at admission in order to avoid placing any additional burden on healthcare practitioners. Furthermore, comorbidities are known to be under-recorded in secondary care,⁵

and, as noted in our manuscript, little comorbidity information was available in the CovidCollab external validation dataset. As you note, ISARIC data was collected prospectively, and comorbidities are therefore likely to be better captured in this dataset, compared to routine/administrative data. Due to these limitations in the use of comorbidities, we therefore considered it of value to explore whether it was possible to develop good prognostic models using only routinely collected and widely available predictors. In a previous study, our team has also found that inclusion of comorbidities was not important in predicting adverse outcomes in hospitalised patients with diabetes.⁶

From the outset, we intended to include imaging data among the candidate predictor variables, as we hypothesised that these were likely to be useful predictors of outcomes, given the nature of the disease. While this was not the case, we still believe it was of value to have evaluated such imaging data, and that it was a strength of our study that we were able to include these and the wide range of other candidate predictors explored in our analysis.

b) Frailty: just to extend this a bit. Frailty is exchangeable with comorbidities in our covid models. An issue with frailty vs comorbidity is that frailty is not validated in younger age groups, which is why we chose not to include it in 4C. Are the authors proposing that frailty be assessed in younger patients so that the score can be used clinically? Is this valid? The capture of frailty scores in administrative datasets is worse than comorbidities, so wouldn't be a good choice from that point of view.

In our UHB derivation dataset, frailty was recorded for 74% of patients. Although this varied slightly by age group, it was relatively well recorded across all age groups in this dataset, as shown in the table below:

Age group	Frailty score recorded	Missing frailty score	Total	% with frailty score
<30	21	14	35	60.0
30-39	30	12	42	71.4
40-49	75	16	91	82.4
50-59	124	22	146	84.9
60-69	138	43	181	76.2
70-79	156	64	220	70.9
80-89	150	64	214	70.1
90+	78	33	111	70.3
	772	268	1040	74.2

We were therefore able to evaluate this as a candidate predictor. While we acknowledge that

frailty score may not be routinely collected in all hospitals for younger patients, where the information is available, it is an important predictor of outcomes (also demonstrated in a recent CovidCollab study)⁷

and we therefore believe its inclusion in the models is justified.

Where not assessed or recorded, frailty is likely to be low, and in practical usage of the model, a 'normal' (low) frailty score can be assumed for missing data.

2) Models.

a) The final models as intended for use are presented in the supplementary tables. I'm struggling with them. Using supplementary table 3 as an example. Age is included as a categorical variable, but the reference level (<30 year) has no events (no one dies). This almost certainly means the model will not have converged, as reflected in the odds ratios for other levels of age (e.g. odds ratio for death of 30-39 y compared with 30 y, OR = 735,275). Confidence intervals are not given (they should be), but likely tend to infinity on the upper bound. Why include a model that has been carefully honed through different approaches that appears to have fundamental mathematical issues, especially when it is easily solved by collapsing the lower levels of the factor? Perhaps I am missing something and apologise if I am.

Thank you for drawing our attention to this issue. The reference category for age has now been amended, and the 50-59 category used instead of <30 years. Similarly, for other variables reference groups have been changed to the 'normal' range where this was not already the case (including for oxygen saturation, pH, WBC, platelets, corrected calcium and eGFR). We would not expect this to make a significant difference to model predictions, because contrasts between populated categories will be stably fit up to the additive constant; this is reflected in the revised model performance values for AUROC and calibration, which are only marginally different for the categorical models with updated reference categories. All tables and results have been updated accordingly. We have also now included the 95% confidence intervals for the model coefficients in all relevant tables.

b) There are clear issues with the categorisation of some of the continuous variables. For instance, for oxygen saturations, patients with levels 80-88% are at greater risk of death than those with saturations <80%. This is not reflected in the continuous GAMs provided in the supplement. Similarly, patients with a pH<7.30-9

7.34 (acidosis) are at lower risk of death than those with a pH in the normal range (there is wiggle in the GAM, although this is uninterpretable given the inclusion of lactate, base excess, and bicarbonate in the same model). Have these models been sense-checked by a clinician? What is the explanation for patients with acidosis or profound hypoxia having a lower risk of death than those who are not? There may be some reason, but I think it more likely that this represents overfitting and these should not have been incorporated into final models. A model describing worse physiological parameters being associated with better survival will have patient safety implications. To aid clarity and interpretation, as noted above, several of the reference categories for these variables have now been updated. The highlighted differences in oxygen saturation and pH are non-significant and may arise due to small sample sizes in certain groups; this is now evident from the overlapping confidence intervals provided for the model coefficients. Furthermore, the coefficients, or differences between the coefficients, are small. All of our results have been reviewed and sense-checked by clinicians.

c) I can't see a table with the ICU admission outcome stratified by other variables (which should be included), but there is presumed to be a similar no/low event rate issue with >90 y.

We did not initially include a second baseline table stratified by ITU admission outcome (secondary outcome) as we felt that the quantity of baseline information was already quite large. However, as requested, we have now created an additional supplementary table

showing the demographic characteristics and comorbidities stratified by those who were and were not admitted to ITU (Supplementary Table 1). You are correct that no patients >90 years were admitted to ITU in the UHB dataset (resulting in a large negative model coefficient and very wide confidence interval).

3) Imputation. It is not clear how imputation was done, I couldn't see the variables that contributed to the models described. Were outcomes included, for instance? Predictive mean matching is described as the method, which would be the case for the continuous missing data, but presumably not for categorical missing data? Most importantly, reference is made to Rubin's rules for combining models, but the table footnotes report "coefficients derived after a single imputation of missing data"? Does that mean the presented final models are not derived using multiple imputation as is described? This is clearly problematic and represents a high risk of bias.

Thank you. We have now clarified the description of multiple imputation in the Methods, Missing data section, as follows: 'Candidate predictors for which >40% of patients had missing data were excluded from the analysis. Further missing continuous variables (vital signs, laboratory tests) and symptoms were imputed using multiple imputation using chained equations (using the R "mice" multiple imputation package). We performed 5 imputations and a maximum of 50 iterations.⁸

Continuous variables were imputed with predictive mean matching, and categorical variables with logistic regression (logreg) or polytomous regression (polyreg). Input variables for the multiple imputation included all available candidate predictor variables in the dataset; outcomes were not included in the imputation variables.'

10

The final presented models and coefficients (using revised reference categories) are derived from the 5 imputed datasets, with coefficients combined using Rubin's rule. The notes previously included beneath the tables to say otherwise have now been removed.

4) Comparison of models: the external validation of the 4C mortality score is greatly appreciated, many thanks. By definition, a new score derived in a particular dataset will perform better than an external score. What would be useful, is to compare both models in an external dataset. Then the added discrimination provided by the extra variables included can be clearly determined. It is unfortunate that a comparison in external validation cohort is not possible, and therefore conclusions around "out performance" need to be tempered by the limitation in methods.

Thank you for the positive feedback. We agree that it would be of value to evaluate both models in an external dataset, and were disappointed not to be able to do so using CovidCollab due to the absence of comorbidity information. We agree that scores will perform better in their derivation dataset than external scores, and have tempered our Discussion and Conclusion accordingly:

Discussion: 'Both the full and reduced UHB-derived models for mortality had slightly better discrimination than the ISARIC 4C score in the UHB data (AUROC 0.753, 95% CI 0.720-0.785). This compares with an AUROC of 0.767 (95% CI 0.760 to 0.773) for the 4C score reported in the original ISARIC validation cohort.⁹

However, better performance may be expected for models evaluated in their development dataset compared to models developed in external datasets. The newly developed UHB model offers an advantage over the ISARIC 4C model in that it utilises only routinely collected patient data recorded at admission, and does not require additional assessment and recording of specific comorbidities (which are often not routinely fully recorded at the point of admission).'

Conclusion: 'The simple, reduced models used only routinely collected data gathered at admission, showed good discrimination and calibration, performed at least as well as the

existing ISARIC 4C score, and performed well in a validation cohort.'

5) XGboost models: the authors make a number of statements around validation and recalibration of the ISARIC XGboost model in their data. I'm a little perplexed about this, as the models are not published. It may be that the authors have used the same variables in their own dataset to derive a gradient boosted tree model and that is what is reported. This though has nothing to do with our own models, as none of the weights are used. All references to this would be best removed.

Yes, as the reviewer observes, we refitted the model variables in our dataset, as opposed to using the original ISARIC model weights. As requested, we have now removed all references to this evaluation, and retained only the external validation of the 4C score in the manuscript, which, as the most widely utilised version of the model, is also the most appropriate and relevant validation to report.

6) Calibration: The 4C mortality model is simple by design and can be improved upon at the expense of added complexity. An important question we and others are
11

looking at is performance in patient subgroups poorly represented in ISARIC, both in terms of discrimination and calibration. It would be great therefore to see an analysis capturing low discrimination and calibration using 4C in, say, different age groups, sex, socioeconomic groups, ethnicities etc. That would help clinicians identify subgroups not well described by the model. This has not been done and it would be great if it could be. Calibration is most useful in the external dataset and including subgroups. It is almost pointless to test calibration in the training dataset. Calibration in the large (intercept) (not reported) and slope should both be reported. The calibration plot in the supplement didn't have a legend in my version so I couldn't quite work out what was what. It wasn't clear how the imputed datasets had been treated in the calibration analysis. This is obviously crucial given the variation that can be seen in calibration across imputed sets.

We agree that evaluation of the 4C score in patient subgroups would be of value; unfortunately, this was not in our pre-specified aims and for many patient subgroups there are unlikely to be sufficient numbers in our dataset to perform this analysis. For our newly developed model, however, we have reported discrimination stratified by both gender and age above/below 60 years (patient numbers were too small to provide more granular age-stratified analysis).

We have reported both calibration slope and intercept for our models in Table 2 (this information had previously been included in a separate Supplementary Table 2, now removed). The calibration plots are combined (using Rubin's rule) across the multiple imputations. This has now been made more explicit in the Methods as follows: 'AUROC and all other metrics (including calibration plots) were combined from all the multiple imputations of the dataset using Rubin's rules for the mean and confidence interval (derived from the standard deviation).'

We apologise that the figure legend for the calibration plot was not visible to you; this was included in the manuscript as follows: 'Figure 1. Calibration plots (observed probability (yaxis) against predicted probability (x-axis)): A. University Hospitals Birmingham (UHB) derivation dataset for mortality outcome; B. UHB derivation dataset for intensive therapy unit (ITU) admission outcome; C. UHB-R derivation/train dataset reduced model for mortality outcome; D. UHB-R derivation/train dataset reduced model for ITU admission outcome; E. CovidCollab external validation dataset reduced model for mortality outcome; F. CovidCollab external validation dataset reduced model for ITU admission outcome.'

7) Time series. From what I understand, multiple logistic regressions models have been created over time and the coefficients for particular variables compared. I like the idea, but it is the stuff of causal inference nightmares. The reader is invited to draw conclusions about the effect of a particular characteristics on the probability of

death on a given day, but this is as likely to reflect the changing population over time (patients die and are excluded) as anything meaningful about the contribution of particular characteristics. I think this would be best removed.

As these are prediction models, no causal interpretation is needed, and as noted by the reviewer, is best avoided. The aim of this analysis was to explore whether a dynamic model would be worth investigating. This is something that our clinician advisory group were
12

interested in, and is a concept which has been evaluated in a non-COVID-19 context in a previous study published in NEJM.¹⁰ We believe the results here are of value in reassuring clinicians that time-dependent effects on prognosis due to effect modification or selection bias in the first week are small. We would therefore prefer to retain this analysis in the report. We acknowledge that there will certainly be changes in the underlying population over the 8-day period, and have therefore avoided inferring too much from this exploratory analysis.

8) 4C deterioration score: Can the authors compare the ITU score with the 4C deterioration score? [https://www.thelancet.com/pdfs/journals/lanres/PIIS2213-2600\(20\)30559-2.pdf](https://www.thelancet.com/pdfs/journals/lanres/PIIS2213-2600(20)30559-2.pdf)

Thank you for this suggestion, and we appreciate that validation of the 4C deterioration score will be of great value. Unfortunately, this was published after our study was undertaken and validation of the score is beyond the original scope of our study. At present, the team do not have the resources to carry out this validation – we hope the reviewer understands. In summary, therefore, I greatly appreciate the work the authors have done here. The final models appear to have significant issues and are not usable. I would be glad to look at a fully revised approach to this. Many thanks again.

Thank you. We hope the above responses, together with the corresponding amendments to the manuscript, address your concerns.

Please do not hesitate to contact me if you require any further information or clarification.

Yours sincerely,

Dr Nicola Adderley, on behalf of all authors

References:

1

Public Health England. Technical summary. Public Health England data series on deaths in people with

COVID-19. 2020. Available at:

https://assets.publishing.service.gov.uk/government/uploads/system/uploads/attachment_data/file/916035/RA_T

Technical_Summary_-_PHE_Data_Series_COVID_19_Deaths_20200812.pdf [accessed 18th June 2021]

2

Public Health England. Coronavirus (COVID-19) in the UK. Deaths in United Kingdom. 2021.

Available at:

<https://coronavirus.data.gov.uk/details/deaths> [accessed 11th June 2021]

3

Aliberti MJR, Covinsky KE, Garcez FB, et al. A fuller picture of COVID-19 prognosis: the added value of

vulnerability measures to predict mortality in hospitalised older adults. *Age Ageing*. 2021 Jan 8;50:32-39.

doi:10.1093/ageing/afaa240

4

White IR, Royston P, Wood AM. Multiple imputation using chained equations: Issues and guidance for

practice. *Statistics in Medicine* 2011;30:377-399.

13

5

Hua-Gen Li M, Hutchinson A, Tacey M, Duke G. Reliability of comorbidity scores derived from administrative data in the tertiary hospital intensive care setting: a cross-sectional study *BMJ Health & Care*

Informatics 2019;26:e000016. doi:10.1136/bmjhci-2019-000016

6

Nirantharakumar K, Hemming K, Narendran P, Marshall T, Coleman JJ. A Prediction Model for Adverse

Outcome in Hospitalized Patients With Diabetes. *Diabetes Care* 2013;36:3566-3572.

doi:10.2337/dc13-0452

7

Geriatric Medicine Research Collaborative. Age and frailty are independently associated with increased

COVID-19 mortality and increased care needs in survivors: results of an international multi-centre study. *Age*

and Ageing 2021;50:617-630. doi:10.1093/ageing/afab026

8

White IR, Royston P, Wood AM. Multiple imputation using chained equations: Issues and guidance for

practice. *Statistics in Medicine* 2011;30:377-399.

9

Knight SR, Ho A, Pius R et al. Risk stratification of patients admitted to hospital with covid-19 using the

ISARIC WHO Clinical Characterisation Protocol: development and validation of the 4C Mortality Score. *BMJ*

2020;370:m3339 doi:10.1136/bmj.m3339

10 Escobar GJ, Liu VX, Schuler A, et al. Automated Identification of Adults at Risk for In-Hospital Clinical

Deterioration. *N Engl J Med* 2020;383:1951-60. doi:10.1056/NEJMsa2001090

VERSION 2 – REVIEW

REVIEWER	Harrison, Ewen University of Edinburgh
REVIEW RETURNED	07-Dec-2021
GENERAL COMMENTS	Many thanks for the opportunity to review this manuscript. I only now have access to the replies to review comments. There are points from the original reviews that remain unaltered, but I have no further comments on manuscript. Many thanks again.